# The Potential of Plum Seed Residue: Unraveling the Effect of Processing on Phytochemical Composition and Bioactive Properties

Sandra Rodríguez-Blázquez [1,2], Laura Pedrera-Cajas [1], Esther Gómez-Mejía [1], David Vicente-Zurdo [1,3], Noelia Rosales-Conrado [1], María Eugenia León-González [1,*], Juan José Rodríguez-Bencomo [1] and Ruben Miranda [2]

[1] Department of Analytical Chemistry, Faculty of Chemistry, Complutense University of Madrid, Avda. Complutense s/n, 28040 Madrid, Spain; sandro08@ucm.es (S.R.-B.); laurpedr@ucm.es (L.P.-C.); egomez03@ucm.es (E.G.-M.); davidvic@ucm.es (D.V.-Z.); nrosales@ucm.es (N.R.-C.); juanjr10@ucm.es (J.J.R.-B.)

[2] Department of Chemical Engineering and Materials, Faculty of Chemistry, Complutense University of Madrid, Avda. Complutense s/n, 28040 Madrid, Spain; rmiranda@ucm.es

[3] Centre for Metabolomics and Bioanalysis (CEMBIO), Department of Chemistry and Biochemistry, Pharmacy Faculty, San Pablo-CEU University, Boadilla del Monte, 28660 Madrid, Spain

* Correspondence: leongon@ucm.es

**Abstract:** Bioactive compounds extracted from plum seeds were identified and quantified, aiming to establish how the brandy manufacturing process affects the properties and possible cascade valorization of seed residues. Extraction with *n*-hexane using Soxhlet has provided oils rich in unsaturated fatty acids (92.24–92.51%), mainly oleic acid (72–75.56%), which is characterized by its heart-healthy properties. The fat extracts also contain tocopherols with antioxidant and anti-inflammatory properties. All the ethanol–water extracts of the defatted seeds contain neochlorogenic acid (90–368 $\mu g \cdot g^{-1}$), chlorogenic acid (36.1–117 $\mu g \cdot g^{-1}$), and protocatechuate (31.8–100 $\mu g \cdot g^{-1}$) that have an impact on bioactive properties such as antimicrobial and antioxidant. Anti-amyloidogenic activity (25 $mg \cdot mL^{-1}$) was observed in the after both fermentation and distillation extract, which may be related to high levels of caffeic acid (64 ± 10 $\mu g \cdot g^{-1}$). The principal component analysis showed that all plum seed oils could have potential applications in the food industry as edible oils or in the cosmetic industry as an active ingredient in anti-aging and anti-stain cosmetics, among others. Furthermore, defatted seeds, after both fermentation and distillation, showed the greatest applicability in the food and nutraceutical industry as a food supplement or as an additive in the design of active packaging.

**Keywords:** bioactive compounds; *Prunus domestica* L. seed; cascade valorization; oils; brandy processing; oxidative stability; antioxidant activity; antimicrobial activity; anti-amyloidogenic activity; nutraceutical application

## 1. Introduction

Currently, there is a growing interest in the agri-food industries for the development of healthier and more nutritious foods, enhancing the emerge of so-called "bioactive compounds". They are known to be natural extra-food components with biological and functional activities, such as antioxidant, anti-inflammatory, anti-diabetic, anti-cancer, anti-viral, and anti-tumor activities, which protect the human body from high levels of free radicals and reactive oxygen species (ROS) related to cell damage [1]. Some of the main bioactive compounds that have aroused the interest of the scientific community are polyphenols, unsaturated fatty acids, and tocopherols, among others [2].

One of the richest sources of bioactive compounds that are poorly studied is *Prunus domestica* L. seed residues [3]. During the production of plum brandy, large quantities of

intact plum stone residues are generated, and although they provide flavor and aromas in the brandy, plum seeds are one of the main by-products that are discarded (15–30 wt.% of the entire stone). This practice, which is usually carried out with plum fruits that are out of specification and cannot be marketed, consists of crushing the plum fruit with a crusher whilst keeping the stone intact, and a subsequent fermentation process in fermentation tanks for 25–30 days to obtain plum brandy with a low alcohol content. Afterwards, it is subjected to a distillation process to separate the alcohol from the aromas and is left to age in stainless steel vessels for a year. Then, the alcohol content is lowered to 42° with water, thus obtaining the plum brandy that ready to be marketed [4]. Bibliographic data indicate that plum seeds undergo changes in their composition during processing to obtain brandy [5]. Therefore, understanding the impact of each stage of plum brandy processing on the remaining seeds, both in terms of the composition of bioactive compounds and their bioactivities, will allow for the designing of efficient valorization, focused on the utter exploitation of the resulting extracts.

According to the nutritional composition, the principal component of these seed by-products from plum brandy processing are oils, which are commonly recovered via Soxhlet extraction with *n*-hexane used as a solvent [6]. Particularly, plum seed oils stand out for their high content of unsaturated fatty acids (UFA), mainly oleic acid (C18:1n9c) (60–73%) and linoleic acid (C18:2n6c) (16–31%) [6–8]. These fatty acids positively contribute to the healthy lipid indexes of the oils (desirable fatty acid (DFA), hypocholesterolemic/hypercholesterolemic (H/H) ratio, atherogenicity (AI), and thrombogenicity (TI)). Moreover, these compounds have an important role in the prevention of cardiovascular diseases, decreasing the concentration of low-density lipoproteins (LDL) that are deposited in the blood vessels, reducing arrhythmia, mortality from coronary heart diseases, and the rate of atherosclerosis, as well as blood pressure [6,9]. Other bioactive components present in the oils are tocopherols, also known as Vitamin E precursors. The predominant tocopherol in plum seed oils is $\gamma$-tocopherol (15.5–16.2 mg·kg$^{-1}$). Notwithstanding, the active form with the greatest functionality in the body is $\alpha$-tocopherol (12.3–3.55 mg·kg$^{-1}$) since it can reduce blood cholesterol levels, as well as coronary mortality or cardiovascular diseases. On the other hand, $\beta$-tocopherol (9.2–11.2 mg·kg$^{-1}$) and $\delta$-tocopherol (2.7–4 mg·kg$^{-1}$) could display anti-inflammatory and antioxidant activities against oxidative stress [10]. Besides tocopherols, fatty acids are also crucial to increase the oxidative stability of oils, thus avoiding the formation of toxic degradation products that reduce their nutritional quality [11].

Nevertheless, after the oil extraction process, a second-generation biowaste is generated, namely defatted plum seeds, which may contain compounds with positive biological activities for the human organism such as phenolic compounds [12]. Among the phenolic compounds found in defatted plum seeds, the main ones were rutin (64–5670 µg·g$^{-1}$), 2,3-dihydroxybenzoic acid (400–3780 µg·g$^{-1}$), gallic acid (143–1890 µg·g$^{-1}$), catechin and epicatechin (1070 and 9.5 µg·g$^{-1}$, respectively), syringic acid (8.7–900 µg·g$^{-1}$), chlorogenic acid (250–300 µg·g$^{-1}$), and caffeic acid (3–300 µg·g$^{-1}$) [13,14]. It has been reported that the presence of these phenolic compounds in defatted plum seed extracts exhibited high antioxidant and neuroprotective activities that inhibit $\beta$-amyloid (A$\beta$) aggregation as well as antimicrobial activities against the growth of pathogens (*Escherichia coli* and *Staphylococcus aureus* among others). This highlights the impetus for the use of these residues as natural functional and nutritional ingredients [3,15–17]. However, *Prunus* seeds, especially those that have been subjected to high temperatures, may present amygdalin, a cyanogenic glucoside recognized by the European Food Safety Authority (EFSA) as an anti-nutritional and toxic compound for the body [18]. Therefore, it is essential to determine the presence of this contaminant, which will also define the potential use of the extracts. And in those extracts that contain this contaminant, it is necessary to apply detoxification processes to ensure its reduction or even its elimination [19].

The high content of bioactive compounds in *P. domestica* L. seeds, with well-known benefits for human health, makes the search for new industrial applications feasible [3].

In contrast, the enrichment of antioxidants in plum seed oils makes them interesting for their use in the food industry as promising substitutes for olive and grape oils. In recent years, plum kernels have gained attention as potential edible cooking oils due to their high oil content [20]. In addition, its outstanding oxidative stability, as well as its antioxidant activity, allows for new applications in the cosmetic industry as an active ingredient in the production of anti-aging, anti-stain, and other potential cosmetics [21]. Not to mention its remarkable heart-healthy indexes, which open new ways for application in the nutraceutical industry for the elaboration of functional foods [22]. On the other hand, phenolic extracts of defatted plum seeds could be of interest to the food and nutraceutical industries as additives in active packaging formulation that protects food from external oxidants, as well as in the pharmaceutical industry as an active ingredient in the elaboration of drugs against neurodegenerative diseases, diabetes, and other important diseases [23–26]. Hence, the antioxidant activity that polyphenols are associated with could justify their use in the cosmetic industry to formulate anti-aging, anti-stain, and sunscreen cosmetics, to name but a few [25,27].

In this line, the present work aims to develop a cascade valorization process of plum seed waste generated in the different stages of plum brandy manufacturing (before fermentation (PBF), after fermentation (PAF), and after both fermentation and distillation (PAFD)) for the obtention of bioactive compounds of special interest such as unsaturated fatty acids, tocopherols, and phenolic compounds. Likewise, the effect of processing on the profile and composition of those compounds in the obtained extracts was evaluated. For this purpose, conventional Soxhlet extraction was used to separate the oils from the defatted seed residues. The resulting oils were characterized in terms of fatty acid profile and their respective heart-healthy indexes, the composition of tocopherols, as well as the evaluation of their antioxidant activities, and the determination of their oxidative stability against lipid degradation. Following this, phenolic compounds were extracted from defatted plum seeds using a matrix solid-phase dispersion (MSPD) methodology, being thereafter characterized in terms of individual and total polyphenol content, antioxidant, anti-amyloidogenic, and antimicrobial activities to explore their potential application in nutraceutical, cosmetic and/or pharmaceutical industries.

## 2. Results and Discussion

### 2.1. Determination of Soxhlet Extraction Yields of Plum Seeds

The oil content of plum seeds (PBF, PAF, and PAFD) obtained during the different stages of plum brandy production was determined using the Soxhlet extraction procedure with *n*-hexane as a solvent [28]. There were considerable differences in the appearance of the oils related to the manufacturing step they proceeded from (Figure S1), with a yellowish tone observed in the PBF and PAF seed oils, contrasting with a dark orange in the PAFD seed oil along with a stronger and more intense aroma. Similarly, significant differences were observed in the weight percentages of plum seed oils and defatted plum seeds. These percentages by weight are shown in Table 1.

**Table 1.** Oils and defatted seeds percentages from plum seed waste.

| Sample | Seed Oil (%, *w*) | Defatted Seed (%, *w*) |
|---|---|---|
| PBF | (37.6 ± 0.4) [a] | (62 ± 1) [a] |
| PAF | (45.2 ± 0.1) [b] | (47 ± 4) [b] |
| PAFD | (68 ± 1) [c] | (30 ± 2) [c] |

Values are expressed as mean ± standard deviation (n = 3) in dry weight. Values followed by a different superscript in a column differ significantly (*p*-value < 0.05), according to one-way analysis of variance (ANOVA) and Fisher's Least significant difference (LSD) test. All data were expressed on a dry basis. Moisture content of initial seeds: plum before fermentation (PBF) (4.5 ± 0.7%), plum after fermentation (PAF) (2.6 ± 0.1%), and plum after both fermentation and distillation (PAFD) (1.9 ± 0.2%). Moisture content of defatted seeds: PBF (4.45 ± 0.07%), PAF (2.4 ± 0.1%), and PAFD (1.82 ± 0.06%).

According to the results in Table 1, significant differences (*p*-value < 0.05) were observed in the contents of each industrial step oil and defatted seed by-products. Regarding the oil content, PBF seed presented the lowest value, withal it was observed that the oil yield increased in the other samples (PAF and PAFD), indicating a direct correlation between the industrial procedure and the extraction itself. However, the opposite behavior was observed when it came to the percentages of defatted plum seeds, the by-products generated in the Soxhlet extraction process. Accordingly, the highest value was achieved in the PBF seed, decreasing during processing and reaching the lowest value in the PAFD kernel. This is attributed to the potential extraction of other hydrophilic components of the seeds during Soxhlet extraction, which enriches the content of defatted seeds. That being so, the lower content of defatted seed in PAFD may be related to its lower content of hydrophilic compounds, such as proteins, sugars, and even polyphenols [12].

Furthermore, these results are in line with data found in the literature. The fat content of plum seeds of the variety *P. domestica* L. extracted by Soxhlet using *n*-hexane as a solvent usually varies between 30 and 38.7%, *w* [6,12]. Concerning the effect of processing on oil content, Rodríguez-Blázquez et al. [12] demonstrated that defatted *P. avium* L. seed residues decreased their content during brandy processing, thereby increasing their oil content.

In this context, both the oil extracted from the seeds and the by-product of defatted seeds could present numerous bioactive compounds with excellent properties or activities, such as polyphenols, tocopherols, and unsaturated fatty acids, among others [6,12,27]. Hence, exploring their nutritional value and potential content has become an emerging requirement for the scientific community.

### 2.2. Evaluation of the Lipid Profile of Plum Seed Oils

#### 2.2.1. Fatty Acid Composition

As previously reported, plum kernel oils are well known for their desirable lipid profile composed mostly of oleic and linoleic acid, with multiple benefits for human health [6,8,17].

In the present work, the fatty acid content of different oils obtained from three plum seed residues (PBF, PAF, and PAFD) was determined to study the effect of processing on the lipid content. The resulting lipid profiles of the plum seed oils are shown in Table 2.

**Table 2.** Fatty acid composition of plum seed oils.

| Fatty Acids | PBF Oil (%) | PAF Oil (%) | PAFD Oil (%) |
|---|---|---|---|
| Palmitic acid (C16:0) | (5.60 ± 0.05) [a] | (5.52 ± 0.01) [a] | (5.15 ± 0.01) [b] |
| Palmitoleic acid (C16:1n7) | (0.765 ± 0.004) [a] | (0.586 ± 0.006) [b] | (0.59 ± 0.03) [b] |
| Margaric acid (C17:0) | (0.0439 ± 0.0008) [a] | (0.04 ± 0.01) [a] | (0.048 ± 0.002) [a] |
| Cis-10-heptadecenoic acid (C17:1n10c) | (0.082 ± 0.004) [a] | (0.092 ± 0.001) [b] | (0.0879 ± 0.0009) [b] |
| Stearic acid (C18:0) | (1.9 ± 0.1) [a,b] | (1.85 ± 0.06) [a] | (2.13 ± 0.04) [b] |
| Oleic acid (C18:1n9c) | (75.56 ± 0.04) [a] | (74 ± 1) [a,b] | (72 ± 1) [b] |
| Linoleic acid (C18:2n6c) | (15.83 ± 0.02) [a] | (18.1 ± 0.9) [a,b] | (20 ± 1) [b] |
| Arachidic acid (C20:0) | (0.176 ± 0.001) [a] | (0.174 ± 0.001) [a,b] | (0.157 ± 0.003) [b] |
| $\sum$ SFA | (7.76 ± 0.07) [a] | (7.58 ± 0.09) [b] | (7.49 ± 0.06) [b] |
| $\sum$ UFA | (92.24 ± 0.07) [a] | (92.42 ± 0.09) [b] | (92.51 ± 0.06) [b] |
| $\sum$ MUFA | (76.40 ± 0.05) [a] | (74 ± 1) [a,b] | (72 ± 1) [b] |
| $\sum$ PUFA | (15.83 ± 0.02) [a] | (18.0 ± 0.9) [a,b] | (20 ± 1) [b] |
| PUFA/SFA ratio | (2.04 ± 0.02) [a] | (2.38 ± 0.09) [a] | (2.7 ± 0.2) [b] |

Results are represented in percentages with estimates of standard deviation (n = 2). Values on the same row with different letters denote significant differences (*p*-value < 0.05) among samples according to ANOVA and Fisher's LSD test. SFA: saturated fatty acids. UFA: unsaturated fatty acids. MUFA: monounsaturated fatty acids. PUFA: polyunsaturated fatty acids (unsaturation $\geq$ 2). PUFA/SFA: ratio between polyunsaturated and saturated fatty acids.

A total of eight fatty acids were identified in the PBF, PAF, and PAFD seed oil samples (Table 2). Plum seed oils, independent of which stage the seeds were collected from during the brandy manufacturing process, were found to be a good source of UFA, mainly

C18:1n9c and, to a lesser extent, C18:2n6c. In addition, considerable amounts of C16:0 and C18:0 were also identified. The remaining four fatty acids were found in small amounts in all samples. These results are comparable with those shown in the literature for plum seeds of the variety *P. domestica* L. Specifically, Rodríguez-Blázquez et al. [6] found that plum seed oil, which was extracted by Soxhlet using *n*-hexane, mainly contain C18:1n9c (72.7 $\pm$ 0.2%) and C18:2n6c (16.4 $\pm$ 0.2%) and, in smaller amounts, C16:0 (5.71 $\pm$ 0.03%) and C18:0 (2.86 $\pm$ 0.05%). Górnás et al. [29] extracted plum seed oils of the same variety with an ultrasound probe using *n*-hexane and reported a composition of C18:1n9c (60.2 $\pm$ 7.4%), C18:2n6c (31.8 $\pm$ 6.9%), C16:0 (5.2 $\pm$ 0.7%), and C18:0 (1.5 $\pm$ 0.4%). Finally, Vladić et al. [8] determined that the lipid profile of *P. domestica* L. seed oils, extracted via supercritical $CO_2$ and cold pressing, was also predominantly C18:1n9c (65–68%), C18:2n6c (22–25%), C16:0 (5.79–5.80%), and C18:0 (1.62–1.92%).

To study the effect of plum seed processing on the lipid content of the seed oil, a one-factor ANOVA statistical study and Fisher's LSD test were carried out (Table 2). PAFD seed oil showed significant differences ($p$-value < 0.05) in all fatty acids, except C17:0 and C18:0, in comparison with PBF seed oil. Moreover, it is noteworthy that C17:0 was the only one that did not present significant differences ($p$-value $\geq$ 0.05) within the process. In addition, several studies affirm that the lipid profile of oils, as well as their bioactive compounds, can be modified with the processing that seeds undergo [5,30]. On one hand, Rabrenović et al. [5] studied the effect of processing that plum seeds of the "Čačanska rodna" variety underwent in obtaining plum brandy and demonstrated that C16:0 content in the obtained cold-pressed oils suffered a significant modification after the distillation procedure in slivovitz brandy manufacturing. The same was observed with the content of SFA, UFA, and MUFA. However, the PUFA content does not differ significantly ($p$-value < 0.05) between the different processed seed residues. Curiously, they found that after distillation, trans-fatty acids begin to appear, such as elaidic acid (C18:1n9t) or linoelaidic acid (C18:2n6t), although in small amounts. It may be related to the isomerization process that C18:1n9c and C18:2n6c undergo with the temperature applied in the distillation process, although the presence of these trans fats heavily reduces the nutritional quality of the oils. On the other hand, Bjelica et al. [30] demonstrated that wine processing of grape seed residues was a significant factor to consider in the lipid content of grape seed residue oils. In this case, they did not observe the appearance of trans-fatty acids and confirmed that the content of C18:1n9c, C18:2n6c, MUFA, and PUFA was modified during the different processing stages.

The presence of high amounts of unsaturated fatty acids, mainly C18:1n9c and C18:2n6c, in all plum seed oils studied means that they can be considered healthy for humans [31]. C18:1n9c is the main MUFA found in the circulatory system and is well known for its many benefits to human health [32]. It is known to be the key energy molecule in the maintenance and development of cell membranes. One of its most characteristic effects is its antioxidant character, as it can regulate both the synthesis and the activities of antioxidant enzymes, even reducing oxidative stress, one of the main hypotheses of Alzheimer's disease [33]. Along with its ability to decrease the expression of proteins related to cholesterol transport, thus reducing cholesterol absorption and oxidation of LDL and preventing atherosclerosis. This fatty acid also has potential anti-cancer activities as it can inhibit the overexpression of oncogenes and their apoptotic effects [32–34]. Regarding C18:2n6c, a major constituent of the dietary PUFA ratio and omega-6 essential fatty acid, it has been highlighted in the scientific community for its many positive activities on human health. Data available in the literature suggest that circulating concentrations of C18:2n6c can reduce the risk of cardiovascular disease not only due to its cholesterol-lowering effect but also its beneficial effect on glucose metabolism. In addition to C18:1n9c, several studies suggest that it may have a beneficial effect on inflammatory parameters, causally correlated with the development of many degenerative diseases, and may also reduce the concentration of LDL, which cause dysrhythmia, coronary heart disease mortality, and blood pressure [35–37]. In addition, the presence of a higher MUFA content compared to PUFA in plum seed oils is potentially interesting as monounsaturated fatty acids are characterized

by increased stability of oils against lipid oxidation, as well as higher anti-inflammatory activities that help to reduce the risk of developing coronary heart disease [6]. In tandem with the high PUFA/SFA ratio of the plum seed oils, which allows for the reduction in cholesterol levels in the blood. Thus, the levels were higher than the recommendations of the UK Department of Health [38], which established the PUFA/SFA ratio in the human diet above 0.45 to present outstanding benefits for the prevention or treatment of cardiovascular diseases.

All things considered, the lipid profile of the plum seed residue oils suggests that these oils may have an interesting value-added potential that needs to be explored; therefore, the determination of healthy quality indexes is necessary to evaluate their possible bioactivities for human health.

### 2.2.2. Heart-Healthy Lipid Quality Indexes

The determination of heart-healthy indexes of oils is an important factor in ensuring their lipid quality [9]. Healthy lipid indexes determined for PBF, PAF, and PAFD seed oils are shown in Table 3.

**Table 3.** Heart-healthy lipid indexes of plum seed oils.

| Lipid Indexes | PBF Oil | PAF Oil | PAFD Oil |
|---|---|---|---|
| Desirable fatty acid (DFA) (%) | (94.18 ± 0.05) [a] | (94.27 ± 0.03) [a] | (94.64 ± 0.01) [c] |
| Atherogenicity (AI) | (0.0607 ± 0.0005) [a] | (0.0597 ± 0.0002) [b] | (0.0557 ± 0.0002) [c] |
| Hypocholesterolemic/Hypercholesterolemic (H/H) | (16.3 ± 0.1) [a] | (16.62 ± 0.05) [b] | (17.8 ± 0.1) [c] |
| Oleic acid/linoleic acid ratio (O/L) | (4.772 ± 0.002) [a] | (4.1 ± 0.3) [a,b] | (3.6 ± 0.3) [b] |

Values are represented as mean ± standard deviation (n = 2). Values with different letters in the same row denote significant differences (*p*-value < 0.05) among samples according to ANOVA and Fisher's LSD test.

According to the DFA index, the formation of two homogeneous groups (*p*-value < 0.05) was observed (one group was formed between PBF and PAF seed oils, and the other group was formed via PAFD seed oil). High values were found in all seed oils (Table 3), evidencing their high hypocholesterolemic properties, and their capacity to reduce blood cholesterol levels in humans. In addition, the H/H ratio is positively correlated with the DFA index and measures the bioactive properties of oils. The values of H/H obtained for plum seed oils were statistically different (*p*-value < 0.05), indicating that this oil quality parameter was modified via processing, and in all studied cases, they were characterized by high values, which confirms that the seed oils studied could reduce the risk of suffering from cardiovascular diseases and could even have bioactive properties such as anti-inflammatory, even managing to reduce the risk of suffering from diabetes [9,39]. Considering the values of the AI index, significant differences (*p*-value < 0.05) were observed in all oils. Furthermore, according to data found in the literature [9], the low levels obtained for this index in all oils could be suitable for the prevention of coronary heart disease. Thus, the lower this value, the healthier the food, indicating the promising potential of these plum seed oils. Finally, the oleic acid/linoleic acid (O/L) ratio was calculated, which is positively correlated with the oxidative stability of the oils [40]. Higher ratios of O/L are associated with a longer shelf life because C18:2n6c (which has two double bonds) is more susceptible to the degradation process than C18:1n9c (which only has one double bond). The oils from plum seed residues obtained in the different stages of plum brandy manufacturing presented high values of the ratio (O/L), and the PAFD seed oil presented statistically significant differences (*p*-value < 0.05) with the PBF seed oil. Therefore, the variation in the O/L ratio in PAFD seed oil may indicate that it is more exposed to oxidative degradation processes [41]. Overall, the lipid indexes and, therefore, the fatty acid content of the plum seed oils were modified during brandy processing. Although PAFD seed oil showed the most suitable heart-healthy indices, namely lower AI and higher DFA and H/H values, its lowest O/L ratio could favor oil degradation and hence the formation of toxic compounds that could drastically

reduce their nutritional quality. In this line, all the studied oils showed heart-healthy lipid indexes that may be of interest as an active ingredient in the food or nutraceutical industry for the prevention of cardiovascular diseases [42]. While PBF and PAF seed oils would be suggested as a potential use in the elaboration of cosmetics against anti-aging or atopic dermatitis, as they present high ratios of O/L, which increase their oxidative stability [40,41].

### 2.3. Determination of Tocopherols Content of Plum Seed Oils

Tocopherols are paramount bioactive constituents of vegetable oils. These compounds are known to be associated with a lower risk of coronary heart disease and cancer. In addition, these compounds prevent lipids and lipid-containing foods from oxidizing during storage, prolonging their stability and shelf life, which is essential when it comes to food production and marketing [43].

The content of tocopherols in plum seed oils before fermentation, after fermentation, and after both fermentation and distillation was determined using a high-performance liquid chromatographic coupled to a photodiode array detector (HPLC/PDA). While the target was the determination of all four vitamin E isoforms (α-, β-, γ-, and δ-tocopherol), this separation was not achieved. Based upon bibliographic research, a stationary phase such as pentafluorophenylsilica (PFPS) was supposedly selective between these slightly different analytes [44], albeit none of them were even retained. Further research landed upon using C18 columns, highlighting the fact that such a stationary phase would never allow the separation between β- and γ-tocopherol due to their similarities in their chemical structures [45]. Abidi et al. [46] suggested the possibility of increasing the separation by using 2-propanol as the organic modifier of the mobile phase, given its lower dissociating effect toward analytes. Nonetheless, the resolution between β- and γ-tocopherol peaks was not acceptable with the mobile phase propanol–water 70:30 (*v/v*), only allowing the detection of high β-tocopherol concentration (which was not expected in plum samples). This fact, together with the broadening of the other chromatographic peaks, which decreased the sensitivity of the method, led to the selection of only methanol as the optimal mobile phase.

Table 4 includes the main analytical characteristics of the chromatographic method developed for the determination of tocopherols. Accordingly, the method was proved to be linear ($R^2$ > 0.9990) in two concentration ranges for all three analytes, other than the β-tocopherol, given its coelution with γ-tocopherol. As far as reproducibility is concerned, it appears that the method developed meets the requirements, particularly for the reproduction of peak areas. The coefficient of variation is less than 5% in all cases. Regarding the retention factor, given the nature of liquid chromatography itself, it is also reproducible, although its coefficient of variation is higher than the former.

**Table 4.** Calibration curve for α-, γ-, and δ-tocopherol.

| Compound | LOD/LOQ (mg·L⁻¹) | Linear Range (mg·L⁻¹) | Calibration Curve | | | CV (%) Intraday (n = 3) | | CV (%) Interday (N = 9) | |
|---|---|---|---|---|---|---|---|---|---|
| | | | a | b (L·mg⁻¹) | R² | k | Area | k | Area |
| α | 0.17 */0.5 ** | 0.5–20 *; 20–100 ** | $(0 \pm 2)\cdot10^3$ *; $(0 \pm 5)\cdot10^4$ ** | $(115 \pm 2)\cdot10^2$ *; $(132 \pm 7)\cdot10^2$ ** | 0.9990 *; 0.9942 ** | 1.39 *; 2.10 **; 0.56 *; 1.20 **; 1.75 *; 2.12 ** | 4.3 *; 2.3 **; 3.7 *; 2.4 **; 3.8 *; 2.5 ** | 6.4 *; 3.9 ** | 3.7 *; 2.6 ** |
| γ | 0.17 */0.5 ** | 0.5–20 *; 20–100 ** | $(3 \pm 4)\cdot10^3$ *; $(0 \pm 5)\cdot10^4$ ** | $(129 \pm 4)\cdot10^2$ *; $(146 \pm 7)\cdot10^2$ ** | 0.9975 *; 0.9957 ** | 1.30 *; 1.79 **; 0.58 *; 1.24 **; 2.42 *; 2.22 ** | 5.2 *; 2.0 **; 3.6 *; 3.2 **; 2.9 *; 2.7 ** | 6.9 *; 4.0 ** | 4.3 *; 2.9 ** |
| δ | 0.10 */0.3 ** | 0.3–20 *; 20–100 ** | $(7 \pm 5)\cdot10^3$ *; $(0 \pm 1)\cdot10^4$ ** | $(88 \pm 4)\cdot10^2$ *; $(117 \pm 2)\cdot10^2$ ** | 0.9923 *; 0.9996 ** | 1.02 *; 1.94 **; 0.50 *; 1.37 **; 2.93 *; 2.45 ** | 3.5 *; 1.9 **; 4.5 *; 3.1 **; 3.3 *; 2.2 ** | 6.7 *; 4.1 ** | 4.1 *; 2.6 ** |

* Analytical parameter for α-tocopherol: 10 mg·L⁻¹, γ-tocopherol: 5 mg·L⁻¹, and δ-tocopherol: 2 mg·L⁻¹.
** Analytical parameter for α-tocopherol: 100 mg·L⁻¹, γ-tocopherol: 50 mg·L⁻¹, and δ-tocopherol: 50 mg·L⁻¹. Limits of detection (LOD) and quantification (LOQ) calculated as 3.3 and 10 times the background noise signal, respectively. All measures were taken at a wavelength of 292 nm.

According to what other authors have stated, the main tocopherol source in plum oil is the γ- homolog [47,48], which, in this study, varied between 5.7 and 11.2 mg·kg$^{-1}$, whereas α- and δ-tocopherol ranged from 2.02 to 2.5 mg·kg$^{-1}$ and 1.48 to 2.2 mg·kg$^{-1}$, respectively. These results are in accordance with those obtained by Popa et al. [47]; in addition, the values are only comparable to a certain extent, considering the scarce information available in the literature about composition changes provoked by processing. The multifactorial ANOVA test performed with the obtained values confirmed that there are significant differences ($p$-value <0.05) between all the analytes in every sample, especially for the γ- homolog, experiencing a great decrease during the fermentation and a noticeable increase after distillation. The α- and δ- homologs present an alike behavior, increasing the concentration during the fermentation process and then maintaining it in the distillation. As can be observed in Figure S2, after fermenting, the appearance of two new peaks within the 5 and 6 min mark is visible. Those two peaks are only present in PAF and PAFD, indicating that during the said procedure, a change is induced in the samples, in addition to the drastic variety in the γ-tocopherol concentration.

As far as the total tocopherol content goes (Figure 1), a decrease in tocopherol content with fermentation can be easily observed, and then, during the distillation process, it increases yet again. Hubert et al. [49] suggested that this may be attributed to the incubation temperature during the fermentation step or the presence of O$_2$ that oxidizes the substrate via Ultraviolet-Visible or heat exposure. Notwithstanding, Winkler-Moser et al. [50] claimed that the total tocopherol content was higher in the post-fermentation samples, despite it being lower in tocopherol content, due to a possible transformation of tocopherols into tocotrienols and vice versa. Not only do tocopherols participate in those antioxidant reactions, but also in "side reactions" that are not fully unraveled yet, in tandem with an interconversion [50]. Bruscatto et al. [51] stated that the accelerated degradation of α-tocopherol resulted in secondary reactions of the tocoferoxil radical with hydroperoxides of fatty acids, which were not oxidized, to form new radicals. However, it may have also reacted with hydroperoxides, forming peroxil radicals and increasing lipid oxidation reactions. The oxidative degradation tocopherols experience is greatly influenced by the oxidation of unsaturated fatty acids, which increases with lipid oxidation, high oxygen concentration and the presence of radicals. It is not possible to only correlate the variation in the concentration to the oxidation reactions, considering it does not explain the increase in γ-tocopherol. Although there is no information concerning how the procedure directly affects the seed, previously, it has been discussed how the presence of alcohol enhances the porosity of the seed, and therefore, it makes it easier for the tocopherols to migrate to the must, which would explain why the concentration of the γ-homolog experiences a great decrease. As the process continues throughout the distillation, the alcohol evaporation has a detrimental effect on the migration, in contrast with the preconcentrating effect for the tocopherols in the seed.

As previously reported, concerning the properties of each homolog, Hensley et al. [52] stated that higher levels of α-tocopherol may induce a decrease in γ, undesirably, due to the capacity of γ-tocopherol to prevent myocardial diseases as well as be less prone to cancer induction than the former. Aksoz et al. [10] indicated that α- and γ-tocopherol have antagonist effects, and a high γ/α concentration ratio implies a higher risk of obesity. In this study, as the β homolog is minoritarian in comparison to γ, it was overruled. Hence, the obtained ratios were (5.0 ± 0.2), (2.27 ± 0.02), and (4.5 ± 0.1) for PBF, PAF, and PAFD, respectively, which implies that the oil with the lower risk is the one obtained from seeds after fermentation, albeit α-tocopherol is observed to reduce the mortality of a heart stroke in contrast with what was observed with γ-tocopherol. Likewise, Seppanen et al. [53] contrasted in a thorough review what other authors have prior observed, landing upon the conclusion that the α-tocopherol is more susceptible to suffer oxidation because of donating its hydroxyl radical than the γ homolog. Therefore, this tocopherol is believed to have higher antioxidant activity, at the very least, at a proven concentration of 40 mg·L$^{-1}$, in contrast with the γ-tocopherol, which requires between 100 and 200 mg·L$^{-1}$ to equate

its capacity. Jung et al. [53] demonstrated that at concentrations higher than 100, 250, and 500 mg·L$^{-1}$ of α-, γ-, and δ-tocopherol, respectively, they acted as pro-oxidants during lipid oxidation, resulting in an increase in the levels of hydroxyperoxide and conjugated dienes. Furthermore, they hypothesized that the higher the concentrations of tocopherols in lipids, the greater the amounts of radical intermediates formed from the oxidation of tocopherols during storage.

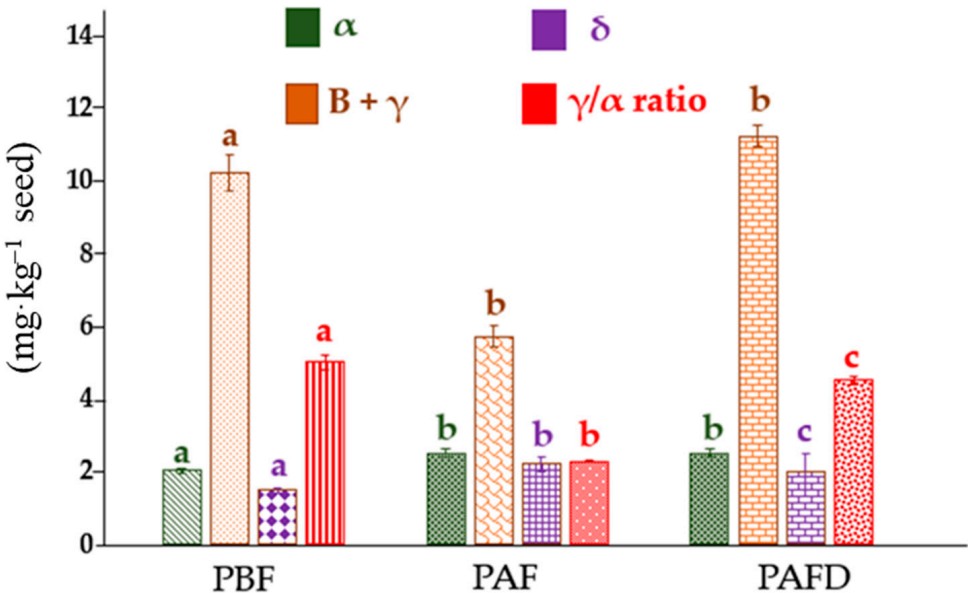

**Figure 1.** Tocopherol content in three plum seed oils. Values are expressed as mean ± standard deviation (n = 4). Different colors represent each tocopherol, in addition to a different pattern and a different letter denote significant differences in their content at a $p < 0.05$ value according to ANOVA and Fisher´s LSD test. PBF: plum before fermentation, PAF: plum after fermentation, and PAFD: plum after both fermentation and distillation. All data are expressed as mean ± standard deviation (n = 4) mg per kg of dry plum initial seed (moisture content of PBF (4.5 ± 0.7%), PAF (2.6 ± 0.1%), and PAFD (1.9 ± 0.2%).

*2.4. Determination of Total Phenolic Content and Individual Polyphenols of Defatted Plum Seeds*

The consumption of phenolic compounds has been shown to have positive health effects, such as preventing a variety of chronic degenerative diseases and delaying the aging process [14]. In this case, seeds obtained from plum kernels that do not meet market specifications were submitted to the correlation analysis between total phenolic content (TPC) and total flavonoid content (TFC) and the effect of plum brandy manufacturing. The TPC and TFC values from phenolic extracts are shown in Figure 2.

According to the TPC results represented in Figure 2, two homogeneous groups (*p*-value < 0.05) were observed. On one hand, PBF seed phenolic extract with PAF extract formed the first group, and on the other hand, PAFD phenolic extract. A statistically significant (*p*-value < 0.05) increase in TPC was observed in PAFD extract ((2.4 ± 0.5) mg GAE·g$^{-1}$ defatted seed) contrasting with the phenolic extracts PBF ((0.63 ± 0.06) mg GAE·g$^{-1}$ defatted seed) and PAF ((0.57 ± 0.08) mg GAE·g$^{-1}$ defatted seed). Accordingly, Sheikh et al. [14] demonstrated that the TPC of *P. domestica* L. seeds subjected to various thermal processes ranged from 0.912 to 0.685 mg GAE·g$^{-1}$. Furthermore, Mehta et al. [16] determined that the TPC of dried *P. domestica* L. seeds were 1.05 mg GAE·g$^{-1}$. The highest TPC (*p*-value < 0.05) was observed in the defatted seed after fermentation and distillation, with respect to the other two defatted seeds, PBF and PAF. This increase may be attributed to the high temperatures to which the plum pits were subjected during the distillation process and could be the result of a simple phenolic compounds reaction, which could lead

to the formation of browning products favored at high temperatures and an increase in polyphenolic content when released [14].

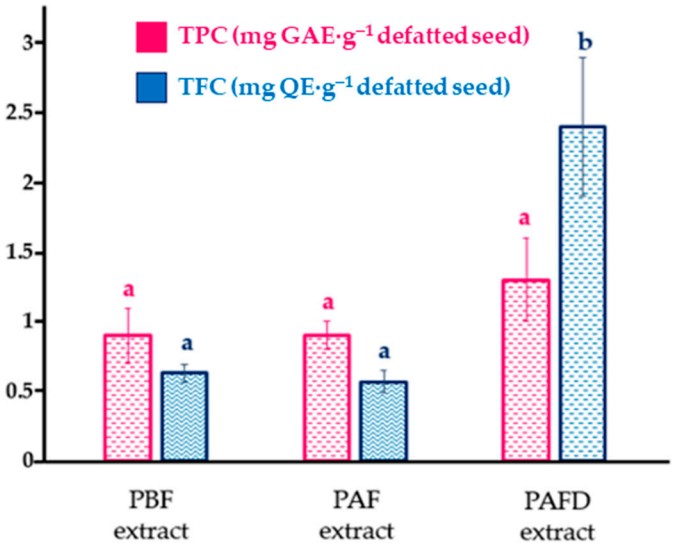

**Figure 2.** Total phenolic content (TPC) (mg GAE·g$^{-1}$ defatted seed) and total flavonoid content (TFC) (mg QE·g$^{-1}$ defatted seed) of plum seed extracts obtained under optimal MSPD extraction conditions. GAE: gallic acid equivalent; QE: quercetin equivalent. All data are expressed on a dry basis (moisture content of PBF defatted seed (4.45 ± 0.07%), PAF defatted seed (2.4 ± 0.1%), and PAFD defatted seed (1.82 ± 0.06%), as mean ± standard deviation (n = 3). Standard calibration curves of TPC: (y = (4.7 ± 0.5)·10$^3$x + (1.7 ± 0.7)·10$^{-3}$, R$^2$ = 0.9992) and of TFC: (y = (3.50 ± 0.0008)·10$^3$x + (14 ± 2)·10$^{-3}$, R$^2$ = 0.9845). Mean values with different letters and with different patterns denote significant differences with *p*-value < 0.05, according to ANOVA and Fisher´s LSD test.

Considering the total flavonoid content, values between 0.9 and 1.3 mg QE·g$^{-1}$ defatted seeds were obtained (Figure 2), which were slightly higher than those found in other studies performed on plum seeds of the European variety (<0.58 mg QE·g$^{-1}$) [16,54]. Likewise, no statistically significant differences (*p*-value < 0.05) were observed between the different plum samples; hence, plum brandy processing does not seem to affect the TFC content.

For the identification and quantification of individual phenolic compounds from three defatted plum seed samples, a non-targeted high-performance liquid chromatography coupled to a quadrupole time-of-flight mass spectrometer (HPLC-ESI-QTOF-MS) methodology was employed. Table S1 shows the retention time obtained for each polyphenol, its respective molecular formula, its molecular pseudo-peak, and its identified fragments. In all cases, a minimum of three fragments were obtained, which allowed for the unequivocal identification of each polyphenol present in the samples. The fragment used for quantification, which corresponded to the most intense fragment in all cases, is indicated in bold. A total of seventeen phenolic compounds were identified. Accordingly, ten phenolic acids, including five hydroxybenzoic acids, were identified as follows: gallic acid (peak 1, RT = 2.7 min), 2,3-dihydroxybenzoic acid (peak 2, RT = 4.9 min), vanillic acid (peak 7, RT = 9.2 min), syringic acid (peak 10, RT = 9.7 min), and protocatechuate (peak 4, RT = 5.0 min), as well as five hydroxycinnamic acids including the following: neochlorogenic acid (peak 3, RT = 4.9 min), chlorogenic acid (peak 6, RT = 7.8 min), caffeic acid (peak 9, RT = 9.6 min), *p*-coumaric acid (peak 11, RT = 11.8 min), and *trans*-ferulic acid (peak 12, RT = 15.2 min). Finally, seven low-molecular-weight flavonoids were identified, including two flavanols: catechin (peak 5, RT = 7.5 min) and its isomer, epicatechin (peak 8, RT = 9.8 min), a flavanone hesperidin (peak 15, RT = 18.4 min), and four flavanols: kaempferol (peak 17, RT = 32.6 min), kaempferol-3-rutinoside (peak 13, RT = 17.7 min), isorhamnetin-3-rutinoside (peak 14, RT = 17.9 min), and quercetin (peak 16, RT = 27.0 min).

The phenolic compounds identified in defatted plum seed extracts are in line with those reported in other studies. As such, Sheikh et al. [14] identified eleven phenolic compounds in plum seeds: gallic acid, chlorogenic acid, catechin, syringic acid, caffeic acid, *trans*-ferulic acid, 2,3-dihydroxy benzoic acid, quercetin, and three new phenolic compounds: rutin trihydrate, ellagic acid, and tannic acid. Savic et al. [13] found rutin, epigallocatechin, gallic acid, *trans*-ferulic acid, syringic acid, epicatechin, caffeic acid, and *p*-coumaric acid. Furthermore, a non-phenolic compound was detected in phenolic extracts from PBF and PAF seeds but not in PAFD. The peak occurred at 6.8 min, with a precursor [M-H] ion at $m/z$ 456.151211 and MS/MS fragments at $m/z$ 391.2956, 323.0798, 263.0537, 221.0551, 161.0341, and 119.0294, resulting in the molecular formula $C_{20}H_{27}NO_{11}$. According to the literature [55], the fragment at $m/z$ 323 was produced because of the neutral loss of a disaccharide [M-H-133], which allowed for the unequivocal identification of this substance known as amygdalin. Amygdalin, a cyanogenic diglucoside compound (D-mandelonitrile-β-D-gentiobioside; syn: D-mandelonitrile-β-D-glucosido-6-β-glucoside), is responsible for the bitterness as well as the toxicity of seeds of the *Prunus* family, including plum seeds [56]. Amygdalin itself is not considered toxic, but its complete hydrolysis generates, in the presence of the enzyme β-glucosidases or α-hydroxynitrilolyases, hydrogen cyanide (HCN), a highly toxic and anti-nutritive compound, and glucose and benzaldehyde [18,56]. Several studies [14] reported that HCN undergoes a degradation process at high temperatures, and accordingly, it has not been found in the defatted plum seed after both fermentation and distillation due to the high temperatures applied during the distillation and subsequent defatting process. Nowadays, it is well known that amygdalin is supposed to be harmful to the human body when levels are exceeded in oral, intramuscular, or intravenous administration [18,56]. The permitted limits set by the European Commission (EU) Regulation (2017/123712) have been particularly a maximum HCN level of 5 mg·kg$^{-1}$ in canned stoned fruits and 35.5 mg·kg$^{-1}$ in alcoholic beverages [57]. In humans, the lethal dose of HCN is considered 50 mg, which is equivalent to 0.8 mg·kg$^{-1}$ body weight. A blood cyanide level of 0.5 mg·L$^{-1}$ is cited in the literature as the threshold for toxicity in humans. Makovi et al. [58] used a similar equipment (HPLC-QTOF-MS) and established the limit of detection (LOD) of amygdalin as 0.015 mg·mL$^{-1}$; thus, as in PAFD seed was not detected, it means that it may contain amygdalin concentrations below that value. Despite the toxicity with which amygdalin is correlated, there is currently a scientific gap, given that several in vitro studies have proven that amygdalin could induce the apoptosis of cancer cells, as well as inhibit their proliferation, which could be potentially interesting in the treatment of neurodegenerative diseases and cancer [59,60]. Be that as it may, amygdalin has even been shown to have potential anti-inflammatory, antinociceptive, and neurotrophic effects [18,55,61,62]. Although its potential benefits are not known with absolute certainty, hence in vivo and clinical studies and safety measures are needed to assess its effects on human health. Nevertheless, it is important to point out that there are still no studies indicating its dermal toxicity levels, and it has been shown that it could have a valuable topical application; therefore, in this line, Gago et al. [63] have shown that an amygdalin analog could reduce the proliferative capacity of psoriasis-stimulated keratinocytes and their inflammatory response in vivo and in vitro. In view of the above, and considering the toxicity of HCN, it would be necessary, firstly, to quantify the former in the extracts obtained from the seed residues before fermentation and after fermentation to ultimately define their suitability as nutraceuticals or agri-food ingredients. Likewise, the application of methodologies to reduce amygdalin levels to minimum values, such as the detoxification processes [14], could be of interest for guaranteeing secure exploitation and applicability of those phenolic seed extracts.

The results corresponding to the phenolic quantification of the three defatted plum seeds evaluated (PBF, PAF, and PAFD) are shown in Table 5. Neochlorogenic acid, chlorogenic acid, protocatechuate, 2,3-dihydroxybenzoic acid, caffeic acid, and quercetin were the main polyphenols found in the three types of samples. Savic et al. [13] reported *trans*-ferulic acid (143 μg·g$^{-1}$), *p*-coumaric acid (118 μg·g$^{-1}$), rutin (64 μg·g$^{-1}$), epigallocatechin

(19 µg·g$^{-1}$), epicatechin (9.5 µg·g$^{-1}$), syringic acid (8.7 µg·g$^{-1}$), gallic acid (6.4 µg·g$^{-1}$), and caffeic acid (3 µg·g$^{-1}$) as major phytochemicals in *P. domestica* L. seeds. Meanwhile, Sheikh et al. [14] found that rutin (5670 µg·g$^{-1}$), 2,3-dihyroxybenzoic acid (3780 µg·g$^{-1}$), gallic acid (1890 µg·g$^{-1}$), catechin (1070 µg·g$^{-1}$), syringic acid (900 µg·g$^{-1}$), caffeic acid (300 µg·g$^{-1}$), and chlorogenic acid (250 µg·g$^{-1}$) were the main polyphenols present in plum seeds. In relation to the latter study, the presence of lower quantities of phenolic compounds in the PBF, PAF, and PAFD seeds of this present study may be due to the plum brandy manufacturing process itself, as well as the subsequent defatting to which they have been subjected, which may favor the degradation or loss of labile phenolic compounds [4,64].

**Table 5.** Quantification of phenolic compounds by HPLC-ESI-QTOF-MS in defatted plum seed extracts.

| Compound | PBF (µg·g$^{-1}$) | PAF (µg·g$^{-1}$) | PAFD (µg·g$^{-1}$) |
|---|---|---|---|
| 2,3-Dihydroxybenzoic acid [A] | 27.3 ± 0.1 [a] | 86 ± 4 [b] | 35 ± 2 [c] |
| Neochlorogenic acid [B] | 217 ± 1 [ab] | 368 ± 98 [b] | 90 ± 3 [a] |
| Chlorogenic acid [B] | 36.1 ± 0.7 [a] | 117 ± 22 [b] | 51 ± 14 [a] |
| Vanillic acid [A] | *n.q.* | 0.9 ± 0.1 [a] | *n.q.* |
| Caffeic acid [B] | 3.6 ± 0.3 [a] | 14 ± 2 [a] | 64 ± 10 [b] |
| Syringic acid [A] | *n.q.* | *n.d.* | *n.q.* |
| *p*-Coumaric acid [C] | 2.3 ± 0.9 [a] | 2.301 ± 0.009 [a] | 6.1 ± 0.3 [b] |
| *trans*-Ferulic acid [D] | 18.5 ± 0.6 [a] | *n.d.* | 3.4 ± 0.2 [b] |
| Kaempferol 3-rutinoside [E] | *n.q.* | *n.d.* | *n.q.* |
| Isorhamnetin 3-rutinoside [F] | 1.86 ± 0.08 [a] | *n.q.* | *n.q.* |
| Quercetin [F] | 15.1 ± 0.7 [a] | 28 ± 2 [a] | 15 ± 8 [a] |
| Kaempferol [E] | 1.38 ± 0.02 [a] | 0.9 ± 0.3 [a] | 19 ± 12 [a] |
| Gallic acid [G] | 8.9 ± 0.1 [a] | 34 ± 2 [b] | *n.q.* |
| Hesperidin [H] | *n.d.* | *n.q.* | *n.q.* |
| Catechin [I] | *n.d.* | *n.d.* | *n.q.* |
| Epicatechin [J] | *n.d.* | *n.d.* | *n.q.* |
| Protocatechuate [A] | 31.80 ± 0.08 [a] | 100 ± 11 [b] | 39 ± 7 [a] |
| Total phenolic acids | 346 ± 1 [a] | 722 ± 123 [b] | 289 ± 12 [a] |
| Total flavonoids | 18.4 ± 0.7 [a] | 29 ± 2 [a] | 34 ± 19 [a] |
| Total phenolics | 364.2 ± 0.7 [a] | 752 ± 121 [b] | 324 ± 0.7 [a] |

Upper-case letters indicate which calibration curve is applied. Standard calibration curves: A—2,3-dyhydroxibenzoic acid ($y = (1.7 \pm 0.1) \cdot 10^7 x + (1.0 \pm 0.1) \cdot 10^6$, $R^2 = 0.9962$, LOD = 1.1 µg·L$^{-1}$ and LOQ = 3.7 µg·L$^{-1}$); B—caffeic acid ($y = (2.98 \pm 0.07) \cdot 10^7 x + (8.0 \pm 0.7) \cdot 10^5$, $R^2 = 0.9990$, LOD = 2.5 µg·L$^{-1}$ and LOQ = 8.3 µg·L$^{-1}$); C—*p*-coumaric acid ($y = (1.7 \pm 0.1) \cdot 10^7 x + (8.0 \pm 0.7) \cdot 10^5$, $R^2 = 0.9964$, LOD = 1.1 µg·L$^{-1}$ and LOQ = 3.7 µg·L$^{-1}$); D—*trans*-ferulic acid ($y = (5.4 \pm 0.1) \cdot 10^6 x + (1.0 \pm 0.1) \cdot 10^5$, $R^2 = 0.9993$, LOD = 1.7 µg·L$^{-1}$ and LOQ = 5.7 µg·L$^{-1}$); E—kaempferol ($y = (3.6 \pm 0.1) \cdot 10^7 x + (1.0 \pm 0.1) \cdot 10^6$, $R^2 = 0.9962$, LOD = 1.4 µg·L$^{-1}$ and LOQ = 4.7 µg·L$^{-1}$); F—quercetin ($y = (3.6 \pm 0.1) \cdot 10^7 x + (3 \pm 1) \cdot 10^6$, $R^2 = 0.9965$, LOD = 1.4 µg·L$^{-1}$, and LOQ = 4.7 µg·L$^{-1}$); G—gallic acid ($y = (2.0 \pm 0.1) \cdot 10^7 x + (1.0 \pm 0.1) \cdot 10^6$, $R^2 = 0.9962$, LOD = 8.6 µg·L$^{-1}$, and LOQ = 28.7 µg·L$^{-1}$); H—hesperidin ($y = (1.37 \pm 0.02) \cdot 10^7 x + (3 \pm 1) \cdot 10^5$, $R^2 = 0.9997$, LOD = 0.3 µg·L$^{-1}$, and LOQ = 0.1 µg·L$^{-1}$); I—catechin ($y = (1.61 \pm 0.01) \cdot 10^7 x + (9 \pm 1) \cdot 10^4$, $R^2 = 0.9998$, LOD = 0.9 µg·L$^{-1}$, and LOQ = 3.0 µg·L$^{-1}$); J—epicatechin ($y = (1.31 \pm 0.02) \cdot 10^7 x + (2 \pm 1) \cdot 10^5$, $R^2 = 0.9996$, LOD = 0.8 µg·L$^{-1}$, and LOQ = 2.7 µg·L$^{-1}$); *n.d.* = not detected; *n.q.* = not quantified. All data are expressed on a dry basis (moisture content of PBF defatted seed (4.45 ± 0.07%), PAF defatted seed (2.4 ± 0.1%), and PAFD defatted seed (1.82 ± 0.06%), as mean ± standard deviation (n = 3). Mean values with different lower-case letters denote significant differences with *p*-value < 0.05, according to ANOVA and Fisher´s LSD test.w

According to the functionalities, the main phenolic compounds determined in the seeds studied were neochlorogenic, chlorogenic, and protocatechuate acids, which are

known to be phenolic compounds with neuroprotective, antibacterial, anti-inflammatory, anti-cancer, anti-diabetic, and hypoallergenic activities [65–68]. 2,3-dihydroxybenzoic acid is attributed to present antibacterial properties against the growth of pathogens such as *Escherichia coli*, *Listeria motocytogenes*, or *Staphylococcus aureus* among others; thus, it is widely used against antibiotic resistance [69]. Caffeic acid, one of the main compounds coming only from the defatted PAFD seed, stands out not only for its effect against oxidative stress but also for its photoprotective effect against ultraviolet radiation, therefore reducing inflammation, genetic mutation, and immunosuppression in human skin cells [70]. In addition, it is key to preventing lipid oxidation and increasing the shelf life of food [71].

Regarding the effect of processing on the phenolic profile of defatted plum seed extracts, three different trends in the variation in phenolic composition with seed processing were observed (Table 5). Firstly, 2,3-dihydroxybenzoic acid, neochlorogenic acid, chlorogenic acid, gallic acid, vanillic acid, protocatechuate, and quercetin showed an increase in their content after the fermentation process (PAF phenolic extract). This could be explained by the fact that during the fermentation process, the appearance of alcohol could increase the porosity of the seeds, allowing the release of these phenolic compounds that, previously to the process, could be found bound to sugars or other biomolecules [72,73]. Other than quercetin, the increase in the content of these phenolic compounds in PAF phenolic extract was statistically significant ($p$-value < 0.05) compared to PBF extract. In addition, once the seeds were subjected to a subsequent distillation process (PAFD), a significant decrease ($p$-value < 0.05) in all the above was observed except for quercetin. According to the literature [74], the high temperatures used in the distillation process favor the degradation of phenolic compounds. Further to this, caffeic acid and $p$-coumaric acid showed a separate behavior, in which their content increased, although not in a statistically significant ($p$-value ≥ 0.05) way after the fermentation process. Nevertheless, after the subsequent distillation process, a statistically significant increase ($p$-value < 0.05) was observed in both cases. As indicated by Bjelica et al. [30], after the distillation process, the porosity of the seeds also increases even more, which could favor the separation of both compounds from the oil and enrichment therefore in the defatted seeds, along with an increase in the content of caffeic acid in PAFD phenolic extract, which could be explained by the fact that high temperatures favor the oxidative degradation process of vanillic acid into caffeic acid [75]. Hence, the content of vanillic acid decreases simultaneously as the content of caffeic acid increases in PAFD extract [72]. Regarding *trans*-ferulic acid and kaempferol, a different trend from those mentioned above was observed. PAF defatted seed extract presents a decrease in the content of both with respect to the initial seed extract (PBF). As indicated by Jericó-Santos et al. [72], this could be explained due to the fact that during the alcoholic fermentation process, a membrane diffusion mechanism of phenolic compounds from the plum seed to the plum must generates a concentration gradient observed by the loss of phenolic compounds in the plum seed and, at the same time, an increase in the content of phenolic compounds in the plum must. Therefore, the decrease in the content of these flavonoids in defatted plum seeds during the brandy production process could be attributed to the migration of phenolic compounds from the seed toward the pulp or other by-products, enriching the brandy obtained in phenolic compounds. Subsequently, after having undergone a later distillation process, a statistically significant ($p$-value < 0.05) increase was observed in both cases and could be correlated with the fact that these phenolic compounds probably were attached to previously glycosylated units and after being subjected to high temperatures they were released, thus heavily increasing their content.

Regarding the total phenolic acids (Table 5), a significant increase ($p$-value < 0.05) was observed in PAF phenolic extract with respect to the initial PBF. After the distillation process, PAFD extract significantly decreased ($p$-value < 0.05) its concentration with respect to PAF phenolic extract. In accordance with other studies that claimed alcohol capacity to increase the porosity of the seed in the fermentation process, which favors the release of phenolic compounds that could be bound to glycosylated units, and after the distillation, the degradation of phenolic compounds is favored [72,74]. This behavior differs from that

observed in the Folin–Ciocalteu spectrophotometric method to determine total polyphenol content. In the latter (Figure 2), it was observed that after both fermentation and distillation processes, the total polyphenol content increased, albeit, considering the low selectivity of the Folin–Ciocalteu reagent, it is likely that it is also reacting with the sugars released in the fermentation process and, therefore, interfering with the measurement [76]. In addition, Folin–Ciocalteu reagent can also react with other interferences, such as aromatic amino acids, proteins, or dehydroascorbic acid, which can affect the precision of the assay. Thus, high temperatures could also favor the formation and condensation of polymeric structures, such as sugars, which can react with the Folin–Ciocalteu reagent [77]. With respect to the total flavonoid content (Table 5), no significant differences ($p$-value $\geq$ 0.05) were observed between the three processes, although an increase is noteworthy, which correlates with that observed with the TFC aluminum complexation colorimetry method.

Overall, it is not only important to determine the phenolic composition of defatted plum seeds, but it is also necessary to evaluate their antioxidant activities to investigate their quality and their possible industrial applications.

### 2.5. Evaluation of Bioactive Properties of Plum Seed Oils and Defatted Plum Seeds

2.5.1. Antioxidant Activity

- Plum seed oils

The determination of the antioxidant capacity of oils is a crucial quality factor as it depends on the composition of bioactive compounds such as unsaturated fatty acids, tocopherols, and others [30]. One of the most popular methods for testing the antioxidant activity of oils is the 2,2-diphenil-1-picrylhydrazil (DPPH) radical scavenging assay [31]. Therefore, this method was used to determine the antioxidant capacity of plum seed oils (PBF, PAF, and PAFD). The antioxidant activity expressed as the concentration of oil required to scavenge 50% of DPPH free radical was represented in Table 6. Trolox was used as standard and its $IC_{50}$ was (($2.5 \pm 0.1) \times 10^{-3}$ mg·mL$^{-1}$). The $IC_{50}$ values of the three oils studied ranged from 20 to 36 mg·mL$^{-1}$. In addition, two statistically homogeneous groups were observed where PAFD seed oil showed significant differences ($p$-value $<$ 0.05) with PBF and PAF oils. The increase in the $IC_{50}$ value and, thus, the decrease in the antioxidant capacity of plum seed oil that has undergone fermentation and subsequent distillation may be due to the high temperatures used in the distillation process, which favor the degradation of bioactive compounds with potential antioxidant capacity [74]. Furthermore, these results could be attributed to the significant decrease in C16:0 content in PAFD seed oil (Table 2), which has been described as a potent antioxidant [78]. Regarding tocopherol content, the only significant difference observed in PAFD oil, with respect to the others, was the increase in β- + γ-tocopherol, given a higher content of the ratio of these tocopherols against the α homolog, which is known to be the active form with high antioxidant properties, could considerably reduce its antioxidant activity.

**Table 6.** Antioxidant activity determined after 24 h storage of plum seed oils.

| Plum Seed Oils | DPPH<br>$IC_{50}$ (mg·mL$^{-1}$ of Oil) |
|:---:|:---:|
| PBF | ($20 \pm 3$) [a] |
| PAF | ($21 \pm 1$) [a] |
| PAFD | ($36 \pm 2$) [b] |

Values are expressed as mean $\pm$ standard deviation (n = 2). Values with different letters denote significant differences ($p$-value $<$ 0.05) among samples according to ANOVA and Fisher's LSD test. $IC_{50}$ values represent the concentration of oil required to scavenge 50% of DPPH free radical.

Notwithstanding, another quality parameter for oils is their stability against lipid oxidation. Hence, the oxidative stability of plum seed oils was measured over a period of 1–22 days using the DPPH free-radical scavenging method. The results obtained are represented in Figure 3. The trend adopted by the $IC_{50}$ data in the studied plum seed oils

(PBF, PAF, and PAFD) was non-linear and fitted adequately to a logarithmic model (with correlation coefficients ($R^2$) between 0.8205 and 0.9727). As a result, a logarithmic model of the first-order degradation kinetic reaction was used to follow the change in antioxidant capacity during storage time. A constant trend was observed in the kinetic curves of PBF and PAF seed oils, which indicated their high stability against lipid oxidation.

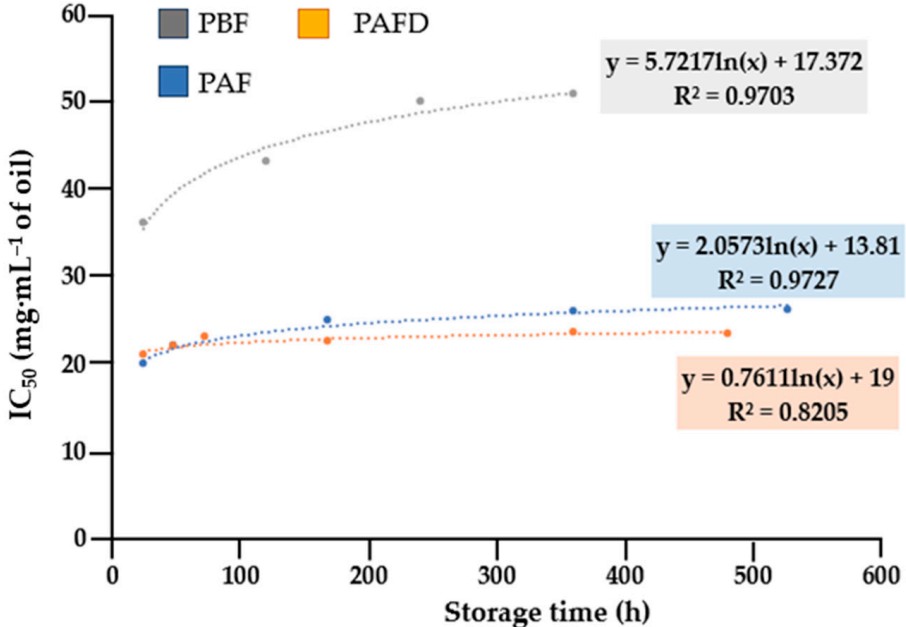

**Figure 3.** Kinetic curves of logarithmic model of $IC_{50}$ values versus storage time of PBF, PAF and PAFD oils. $IC_{50}$ values represents the concentration of oil required to scavenge 50% of DPPH free radical.

However, a different trend was observed in PAFD seed oil, where a progressive increase in $IC_{50}$ could be observed with storage time, which showed that this oil was more susceptible to lipid oxidation.

On the other hand, the half-life time ($t_{1/2}$) was calculated for each oil following the procedure indicated by Rodríguez-Blázquez et al. [6]. The parameters corresponding to the linear fit (intercept, slope, and correlation coefficient ($R^2$)) are represented in Table 7.

**Table 7.** Kinetics parameters corresponding to the linear fit (intercept, slope, and correlation coefficient ($R^2$)) of $\ln_{c.antioxidants}$ versus storage time of plum kernel oils.

| Plum Seed Oils | Intercept (mg·mL$^{-1}$) | Slope (h$^{-1}$) | $t_{1/2}$ (h) | $R^2$ |
|---|---|---|---|---|
| PBF | $(-4.2 \pm 0.1) \times 10^{-3}$ | $(4.05 \pm 0.04)$ | 1732 (72 days) | 0.78 |
| PAF | $(-2.0 \pm 0.5) \times 10^{-4}$ | $(4.03 \pm 0.01)$ | 3465 (144 days) | 0.64 |
| PAFD | $(-9 \pm 3) \times 10^{-3}$ | $(3.7 \pm 0.2)$ | 81 (3 days) | 0.86 |

A linear trend in the data was observed in all plum seed (PBF, PAF, and PAFD) oils. The slope refers to the first-order constant (k) and the intercept to the logarithm of the initial concentration of antioxidant compounds ($\ln_{c0}$).

According to the results indicated in Table 7, it was observed that all plum seed oils conformed to first-order kinetics with a high correlation factor ($R^2$) ranging from 0.64 to 0.86. The half-life time ($t_{1/2}$) in which the oils are considered stable was different for each of them. The longest shelf life was observed for PAF seed oil. PBF seed oil presented a high half-life of 1732 h (72 days); however, a drastic decrease in this parameter was observed in PAFD seed oil. Hence, the results obtained may indicate that the oil composition significantly affects their antioxidant activity and, thus, the oxidative stability [41]. A crucial factor in the oxidative stability of oils is the O/L ratio [79]. That being so, PAFD seed oil had the

lowest proportion of oleic acid with respect to linoleic acid (3.6 ± 0.3), which could explain its lower stability against lipid oxidation since linoleic acid (which has two double bonds) is more susceptible to degradation than oleic acid (which only has one double bond). In addition to the fatty acid content of the oils, other bioactive compounds that influence oxidative stability are tocopherols. According to the tocopherol content shown in Figure 1, PAFD seed oil differed from the others in the significant increase in the content of β- + γ-tocopherol, so a greater proportion of these tocopherols compared to the α homolog could result in a detrimental effect on oxidative stability [80].

According to the results obtained, it seems that the fatty acid content was the main variability factor in the oxidative stability of the oils, yet it is necessary to further study the possible correlations of tocopherols and fatty acids in the lipid fraction to fully prove it. Overall, PBF and PAF oils, due to their high stability against lipid degradation and their relevant antioxidant activity, could present potential applications in the cosmetic industry as an active ingredient in the production of anti-aging or other potential cosmetics [81]. However, in the case of PAFD oil, because of the low oxidative stability compared to PBF and PAF oils, the implementation of natural stabilizing agents is required to increase their stability [82].

- Defatted seed phenolic extracts:

Concerning PBF, PAF, and PAFD phenolic extracts, the antioxidant capacity was measured using two different methods: the DPPH -assay and the in vitro thiobarbituric acid reactive substances (TBARS) tissue-based assay, using the porcine brain as a biological substrate to evaluate the inhibition of lipid peroxidation. This assay was developed to estimate the oxidative damage caused to lipid membranes by oxidizing agents via the degradation product, malondialdehyde (MDA), which reacts with thiobarbituric acid (TBA) to form MDA-TBA$_2$ adducts [77]. The results obtained for the antioxidant activity of defatted plum seeds are shown in Table 8.

**Table 8.** Antioxidant capacity (DPPH free-radical scavenging assay and TBARS assay) of defatted plum seed phenolic extracts.

| Defatted Seed Phenolic Extracts | DPPH IC$_{50}$ (mg·g$^{-1}$) | TBARS IC$_{50}$ (mg·g$^{-1}$) |
| --- | --- | --- |
| PBF | (1.0 ± 0.2) [a] | (2.1 ± 0.1) [b] |
| PAF | (0.9 ± 0.2) [a] | (5.0 ± 0.6) [b] |
| PAFD | (1.9 ± 0.1) [b] | (1.3 ± 0.1) [a] |

Values are represented as mean ± standard deviation (n = 3) on a dry basis (moisture content of PBF defatted seed (4.45 ± 0.07%), PAF defatted seed (2.4 ± 0.1%), and PAFD defatted seed (1.82 ± 0.06%). Values with different letters denote significant differences ($p$-value < 0.05) among samples according to ANOVA and Fisher's LSD test. IC$_{50}$ values represent the concentration of defatted phenolic extracts sample required to scavenge 50% of DPPH free radical.

All the defatted seed phenolic extracts studied (Table 8) presented adequate antioxidant activity (DPPH IC$_{50}$ of 0.9–1.9 mg·g$^{-1}$ and TBARS IC$_{50}$ of 1.3–5.0 mg·g$^{-1}$). The positive control, Trolox, presented a value of IC$_{50}$ = 2.5 ± 0.1 mg·mL$^{-1}$ for the DPPH antioxidant activity assay and a value of IC$_{50}$ = 11.6 ± 0.2 mg·mL$^{-1}$ for the TBARS assay. In addition, the Trolox positive control showed a low coefficient of variation for DPPH (4%) and TBARS assays (2%), showing a high reproducibility in the applied methods. In the DPPH and TBARS assays, two statistically homogeneous groups ($p$-value < 0.05) were obtained. In the DPPH trial, PBF and PAF phenolic extracts showed significant differences ($p$-value < 0.05) with PAFD phenolic extracts, while in the TBARS assay, PAF phenolic extract showed statistically significant differences ($p$-value < 0.05) with respect to the other two phenolic extracts. Considering both assays, two different trends were observed. In the case of the DPPH free-radical scavenging assay, the antioxidant capacity in PAF extract increased, which resulted in a lower IC$_{50}$ value, contrary to what was obtained after the consecutive distillation process (PAFD), where the antioxidant capacity decreased with respect

to PBF phenolic extract. Furthermore, according to the results shown in Table 5, this radical antioxidant capacity may be influenced by variations in the content of 2,3-dihydroxybenzoic acid, neochlorogenic acid, chlorogenic acid, gallic acid, vanillic acid, protocatechuic acid and quercetin in the defatted seed extracts under study. The TBARS anti-lipid peroxidation assay showed that after fermentation, the antioxidant capacity decreased, contrasting with an increment that occurred after distillation, compared to the PBF extract. Considering the phenolic content results shown in Table 5, TBARS antioxidant activity could be influenced by the variations in the phenolic content: caffeic acid, *p*-coumaric acid, and kaempferol.

In accordance with the antioxidant results presented in Table 8, all plum seed phenolic extracts presented outstanding lipid peroxidation inhibition activity, being useful as active ingredients in the development of active packaging to protect foods from external oxidizing agents and, thus, prolonging their shelf life [25]. In addition, all phenolic extracts showed high DPPH antiradical activity, which may be useful in the cosmetic industry as active ingredients in the development of anti-aging cosmetics, anti-stain, or even sunscreens [27]. Yet, the presence of amygdalin in PBF and PAF defatted seeds limits their application, requiring an amygdalin detoxification process to reduce its levels and, subsequently, security trials to ensure their applicability in the industrial sectors [18].

### 2.5.2. Neuroprotective Activity

Another bioactivity to consider is the neuroprotective one. Several studies [83,84] demonstrated that polyphenols can interact with A$\beta$ peptides and form the polyphenol-protein A$\beta$ interaction that blocks the self-association of A$\beta_{42}$ monomers to produce low-molecular-weight oligomers. A$\beta_{42}$ plaques are known as the main neuropathology of Alzheimer´s disease, being the keystone of the "amyloid cascade hypothesis".

Therefore, neuroprotection offered by defatted plum seed phenolic extracts after both fermentation and distillation was evaluated as the ability to inhibit A$\beta_{42}$ induced aggregation. This specific extract was chosen because it was the only one in which the presence of amygdalin had not been detected. Under normal conditions, in the absence of this neurodegenerative disease, A$\beta_{42}$ protein fibrils remained disaggregated, as shown in Figure 4a. However, in metal-induced Alzheimer´s disease, A$\beta_{42}$ fibrils formed aggregates often called amyloid plaques. This increase in amyloid aggregation can be observed in Figure 4b, where co-incubation with Fe(II) induced fibrils to form a dense tangle. Curcumin, used as a negative control due to its well-known anti-amyloidogenic properties, inhibited amyloid aggregation induced by Fe(II) (Figure 4c). Likewise, to evaluate the neuroprotective effect of PAFD phenolic extract, two different concentrations were tested (25 and 50 mg·L$^{-1}$). In the absence of metal, no increase in amyloid aggregation was observed, making PAFD seed extract non-neurotoxic under these conditions (as shown for 25 mg·L$^{-1}$ in Figure 4d). However, the highest concentration (50 mg·L$^{-1}$) was not able to totally inhibit metal-induced aggregation (Figure 4f), although a slight improvement was observed in comparison to Fe(II)-induced aggregation. On the other hand, the lowest plum extract concentration (25 mg·L$^{-1}$) could totally inhibit A$\beta_{42}$ aggregation (Figure 4e), providing a similar pattern to the obtained with curcumin, therefore being the most effective concentration for the treatment of this condition. The optimum concentration of phenolic extract that allowed maximum inhibition of metal-induced and A$\beta_{42}$ aggregation was comparable to that obtained by Gómez-Mejía et al. [85] for grape (*Vitis vinifera* L.) seed extracts, where they observed a similar effect at concentrations of 15 and 29 mg·L$^{-1}$ phenolic extract.

Regarding fibril morphology, in normal conditions, the average width was $(9 \pm 2)$ nm, which slightly reduced in the presence of plum extract to $(6.5 \pm 0.6)$ nm (for 25 mg·L$^{-1}$) and $(6 \pm 1)$ nm (for 50 mg·L$^{-1}$). However, an increase up to $(13 \pm 2)$ nm was measured in the presence of Fe(II), which could be reduced to $(6 \pm 1)$ nm via co-incubation with curcumin. Similar measures were observed in the case of Fe(II) with plum extract at 25 mg·L$^{-1}$ $((6.1 \pm 0.6)$ nm), while the negative effect of 50 mg·L$^{-1}$ extract was also confirmed under this parameter, achieving widths of $(11 \pm 2)$ nm, statistically similar to Fe(II) ones.

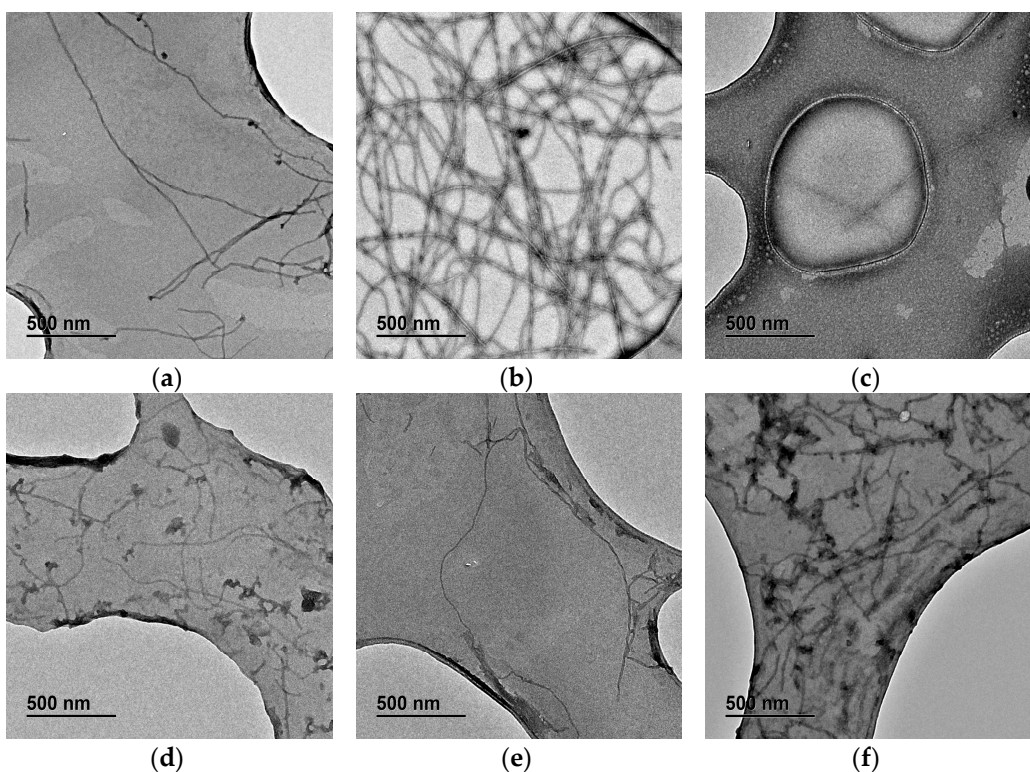

**Figure 4.** Transmission electron microscopy micrographs of $A\beta_{42}$ fibers under the effect of plum extract at different concentrations. (**a**) $A\beta_{42}$ alone, (**b**) $A\beta_{42}$ + Fe(II), (**c**) $A\beta_{42}$ + Fe(II) + curcumin, (**d**) $A\beta_{42}$ + 25 mg·L$^{-1}$ of extract, (**e**) $A\beta_{42}$ + Fe(II) + 25 mg·L$^{-1}$ of extract, (**f**) $A\beta_{42}$ + Fe(II) + 50 mg·L$^{-1}$ of extract. Concentration of: $A\beta_{42}$= 50 μM; Fe(II)= 50 μM; Curcumin = 50 μM.

As a whole, it can be clearly stated that PAFD phenolic extract inhibited $A\beta_{42}$ protein aggregation and minimized the fibrils width, including that induced by metals, which is highly enhanced, highlighting its potential as a nutraceutical ingredient with neuroprotective properties [85].

2.5.3. Antimicrobial Activity

Antimicrobial activity is considered another interesting bioactive property of phenolic compounds in the development of food packaging applications or agri-food ingredients [25]. The results of the antimicrobial activity against two ubiquitous foodborne pathogens are presented in Table 9. As such, the lowest minimal inhibitory concentration values against *Escherichia coli* Gram-negative bacteria were observed for PAF and PAFD phenolic extracts (minimal inhibitory concentrations (MIC) = 20 mg·mL$^{-1}$). However, PBF phenolic extract required concentrations above 20 mg·mL$^{-1}$ to inhibit bacterial growth. With respect to the positive control streptomycin, the MIC against *E. coli* was obtained for a concentration of 20 mg·mL$^{-1}$, which showed the appropriate antimicrobial activity of the extracts. Regarding the *Staphylococcus aureus* Gram-positive bacteria, only PAFD phenolic extract displayed antimicrobial potential, with a MIC value of 20 mg·mL$^{-1}$. A more effective concentration was achieved with streptomycin standard (2.5 mg·mL$^{-1}$), which showed its effectiveness in inhibiting bacterial growth. The differences in the antimicrobial activities of each of the phenolic extracts may be correlated with the difference in phenolic composition, hence the necessity to study their correlations via more advanced studies. In addition, the antimicrobial activities obtained for plum seed phenolic extracts agree with those reported by Alam et al. [15]. Thus, according to the antimicrobial results of the defatted plum seed extracts shown in Table 9, PAFD phenolic extract could be applicable as an added ingredient in the design of active packaging to prevent the growth of *Staphylococcus aureus* in food [25]. Similarly, PAF and PAFD phenolic extracts could have the same applicability

in inhibiting the proliferation of *Escherichia coli* bacteria in food. Once again, in the case of PAF phenolic extract, it is crucial to apply detoxification methods as well as carry out safety tests to ensure its unhazardousness.

**Table 9.** Antimicrobial activity of plum seed phenolic extracts.

| Phenolic Extracts | Gram-Negative Bacteria *Escherichia coli* MIC (mg·mL$^{-1}$) | Gram-Positive Bacteria *Staphylococcus aureus* MIC (mg·mL$^{-1}$) |
|---|---|---|
| PBF | >20 | >20 |
| PAF | 20 | >20 |
| PAFD | 20 | 20 |
| Streptomycin antibiotic-positive standard | 20 | 2.5 |

Values are represented as mean ± standard deviation (n = 3). MIC means minimal inhibitory concentration.

### 2.6. Multivariate Statistical Analysis

With the aim to correlate the composition of plum seed oils and defatted seed phenolic extracts with their bioactive activities, a multivariate statistical analysis by principal component analysis (PCA) was carried out.

For plum seed oils, correlations between total fat content, fatty acid, and tocopherol content with DPPH free-radical scavenging activity and oil oxidative stability were studied. In addition, the processing effect upon the plum seeds was investigated. The resulting PCA has allowed the reduction in fourteen studied experimental factors to two principal components (PC) that explained 100% of the total data variability (Figure 5). PBF seed oil was characterized by the highest C16:1n7 content, while PAFD oil was characterized by the highest C18:0 content. PAF oil was depicted by the maximum C17:1n7c content and the greatest stability against lipid oxidation. With respect to the oil antioxidant activity, a positive correlation was observed between DPPH antioxidant capacity (low level of IC$_{50}$) and C16:0, as well as C20:0 content, and to a lesser extent, with C18:1n9c content. This correlates with what was observed by Rodríguez-Blázquez et al. [6] for *Prunus* seed oils. A high negative correlation was established between the oxidative stability and C18:0 content, in tandem with a negative correlation between β- + γ-tocopherol content and oxidative stability. All this indicated that PAFD oil had the lowest oxidative stability due to a higher content of β- + γ-tocopherol and C18:0.

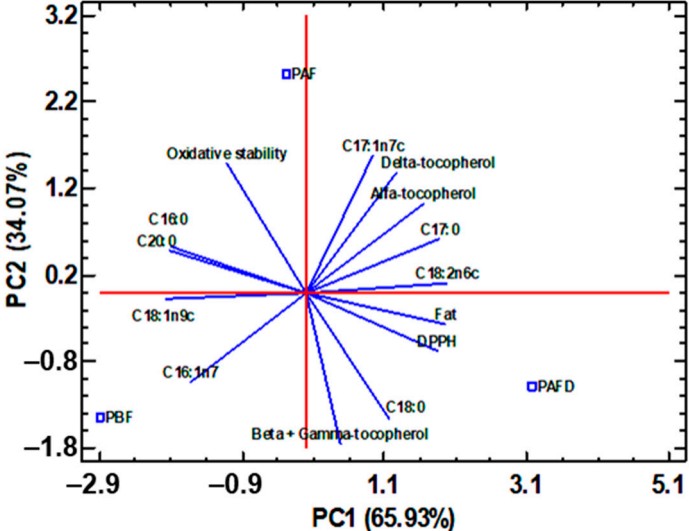

**Figure 5.** Two-dimensional principal component analysis plot of scores (PBF, PAF, and PAFD seed oils ) and loadings (fat content, fatty acid content, tocopherol composition, antioxidant activity (IC$_{50}$ value), and oxidative stability of plum seed oils).

Considering everything mentioned above, the PAF seed oil with the highest content of C17:1n7c could have potential applicability as an active ingredient in the preparation of drugs against cardiovascular diseases since this MUFA can reduce total and LDL cholesterol levels. In relation to carbohydrates, they increase high-density lipoprotein (HDL) cholesterol levels and thus decrease plasma triglyceride levels [86]. Not only its greater oxidative stability but also its high antioxidant activity allows its use in the production of anti-aging cosmetics, among others. PBF seed oil could also be of interest in the production of cosmetics as well as in the pharmaceutical industry as a treatment against cardiovascular diseases due to the high levels of C18:1n9c and C16:1n7 found [6,81]. Both PBF and PAF seed oils are the most interesting for their applicability in the nutraceutical, cosmetic, and food industries since PAFD oil is the one with the least oxidative stability, requiring natural stabilizers that increase its useful life. Moreover, its high content of C18:0, a saturated fatty acid, is known for increasing blood cholesterol levels with negative effects on health.

On the other hand, for plum seed phenolic extracts, correlations between individual polyphenols, TPC, and TFC, as well as the different bioactive properties (antioxidant and antimicrobial activities), were considered in the PCA plot in Figure 6. Vanillic acid, syringic acid, kaempferol-3-rutinoside, isoharmentin-3-rutinoside, hesperidin, catechin, and epicatechin were not included in the PCA study due to their low concentration in the samples, close to the limit of detection and the limit quantification of the method, which makes its contribution to the multivariate study irrelevant or minimal.

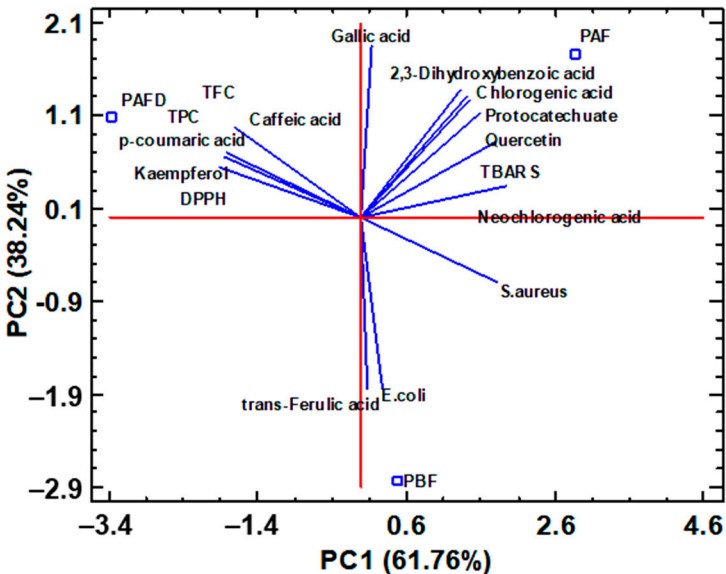

**Figure 6.** Principal component analysis biplot of scores (PBF, PAF and PAFD plum seed phenolic extracts) and loadings (individual polyphenols, TPC, TFC, thiobarbituric acid reactive substances (TBARS) and DPPH antioxidant assays and antimicrobial activities against *Escherichia coli* and *Staphylococcus aureus*).

According to the PCA biplot represented in Figure 6, It was observed that PBF defatted seed phenolic extract was characterized by the highest content of *trans*-ferulic acid, while PAF extract presented high levels of 2,3-dihydroxybenzoic acid, neochlorogenic acid, chlorogenic acid, gallic acid, vanillic acid, protocatechuate, and quercetin. In addition, these kinds of polyphenols were correlated by the greater antioxidant capacity DPPH (lower $IC_{50}$). For its part, PAFD phenolic extract was characterized by high levels of *p*-coumaric acid, kaempferol, which contributes to the TFC, as well as high content of caffeic acid, which seems to have the greatest effect on TPC. In addition, the latter extract presented the highest antioxidant capacity, TBARS, so these polyphenols contribute to increasing the shelf life of foods [25]. This residue was also positively correlated with the greatest antimicrobial activity with respect to *Staphylococcus aureus* (lowest MIC) and, to a lesser extent, with

*Escherichia coli*. Regarding the phenolic profile involved in antimicrobial activity, it was observed that gallic acid had the highest activity against the growth of *Escherichia coli*, while caffeic acid had the highest activity against the growth of *Staphylococcus aureus*. According to literature data, gallic acid is well known to have outstanding activity against the growth of the *Escherichia coli* pathogen, so much so that a high content of this polyphenol in phenolic extracts could be of interest [87]. Not to mention caffeic acid, which is also recognized for its inhibitory work on the growth of *Staphylococcus aureus*, hence has potential application in the design of active packaging against the growth of this pathogen in food, thus increasing its shelf life [25,88].

To summarize everything that has been already mentioned, PAFD phenolic extract would be the most interesting to be applied in the food and nutraceutical industry since it is the only one where amygdalin was not detected and thus could be included as an active ingredient in the production of active packaging against *Escherichia coli* and *Staphylococcus aureus* [25]. Furthermore, the presence of *p*-coumaric acid with kaempferol could highlight its applicability as a dietary supplement against cardiovascular diseases or even in the cosmetic industry in the production of cosmetics that reduce the development of erythema and skin pigmentation due to exposure to ultraviolet radiation [89,90]. Withal, the other two defatted seed extracts cannot be left behind since PAF and PBF phenolic extracts presented the greatest DPPH antioxidant activity and high levels of trans-ferulic acid, which may increase the protection against oxidative stress, respectively. Thus, a possible applicability for these extracts may be the nutraceutical industry once the amygdalin is removed [27,91].

## 3. Materials and Methods

### 3.1. Reagents, Standards, Bacterial Strains, and Solvents

Analytical grade reagents were required in the experimental procedures: *n*-Hexane (96%), methanol (MeOH, ≥99%), and ethanol absolute (EtOH, ≥99.8) for HPLC gradient quality; acetonitrile (ACN) and formic acid (FA) of MS quality. Hydrochloric acid (HCl, 37%) and sulfuric acid ($H_2SO_4$, 95–97%) were supplied by Scharlab (Barcelona, Spain). Dimethyl sulfoxide (DMSO, ≥99.9%), 2N Folin–Ciocalteu reagent, 2,2-diphenyl-1-picrylhydrazyl (DPPH, ≥99.9%), sodium methoxide (95%), trichloroacetic acid (TCA, 99%), thiobarbituric acid (≥98%), sanitary ethanol (96%), 1,1,1,3,3,3-hexafluoro-2-propanol (HFIP), curcumin (>65%), and LB broth with agar and Tryptic Soy Broth (TSB) were all supplied by Sigma-Aldrich (St. Louis, MO, USA). Curcumin was stored at −20 °C prior to analysis. Amyloid β protein fragment 1–42 ($A\beta_{1-42}$) (A9810, CAS: 107761-42-2, MW 4514.04 g mol$^{-1}$) was also obtained from Sigma-Aldrich (St. Louis, MO, USA) for incubation assays and stored at −80 °C until analysis. The standard Trolox (≥97%) was provided by Sigma-Aldrich (Burghasen, Germany). 2-Propanol for HPLC gradient ($C_3H_8O$, PrOH), Aluminum chloride 6-hydrate (99%), sodium carbonate anhydrous, sodium hydroxide (NaOH, 98%), *p*-iodonitrotetrazolium chloride (INT, 98%), ferrous-sulfate heptahydrate ($FeSO_4·7H_2O$, 98%), and sodium nitrite (≥98%) were obtained from Panreac (Barcelona, Spain). Silica was purchased from Fisher Scientific (Pittsburgh, PA, USA).

Fatty acid standards FAME 37 component SUPELCO Ref CRM47885, tridecanoic acid (C13:0, ≥99%), and ethyl nonanoate (>98%) were purchased from Sigma-Aldrich (Barcelona, Spain).

α-tocopherol (≥96%), γ-tocopherol (≥96%), and δ-tocopherol (≥90%) standards were purchased from Sigma-Aldrich (St. Louis, MO, USA). β-tocopherol stock solution was supplied by Sigma-Aldrich (St. Louis, MO, USA) and was already dissolved in methanol, and the remaining stock solutions of 1000 mg·L$^{-1}$ were prepared, dissolving the adequate quantity of analyte in methanol as well. Those solutions were later further diluted for the purpose of a calibration curve into solutions of 0.3, 0.5, 2.0, 5.0, 10.0, 20.0, 50.0, 75.0, and 100 mg·L$^{-1}$. Stock solution stability was also considered, and it was established in three months, at the very least, via conservation in darkness in the freezer at −20 °C.

Phenolic standards gallic acid monohydrate (≥98.0%), chlorogenic acid (≥95.0%), dihydroxybenzoic acid (≥97.0%), caffeic acid (≥98.0%), catechin (≥98.0%), *p*-coumaric

acid (≥98.0%), epicatechin (≥98.0%), rutin trihydrate (≥95.0%), *trans*-ferulic acid (98%), myricetin (≥96.0%), resveratrol (≥99.0%), quercetin (≥95.0%), kaempferol (≥97.0%), and naringin (≥95.0%) were obtained from Sigma-Aldrich (St. Louis, MO, USA). Hesperidin (≥98.0%) was provided by European Pharmacopoeia. Phenolic stock solutions (200 mg·L$^{-1}$) were prepared in MeOH, an ethanol–water mixture 80:20 (*v/v*) (quercetin), or a 5% (*v/v*) DMSO aqueous solution (hesperidin). They were stored in the dark at 4 °C or at −80 °C (hesperidin, *trans*-ferulic, myricetin, and caffeic acid) for up to one month. Fresh working standard solutions were prepared daily by diluting stock solutions as required. Nylon membrane filters with 0.22 μm pore size (Teknokroma, Barcelona, Spain) were used for mobile phase filtration before chromatographic analysis.

The antimicrobial activity of the extracts was evaluated against Gram-positive bacteria *Staphylococcus aureus* (ATCC 29213) and Gram-negative bacteria *Escherichia coli* (ATCC 25922). Streptomycin sulfate salt was used as a positive standard and was provided by Sigma-Aldrich (St. Louis, MO, USA).

### 3.2. Raw Materials

Plum stones of the family *P. domestica* L. and the variety of "Queen Claudia" before fermentation (PBF), as well as those obtained after different stages of plum brandy production, in particular stones after fermentation (PAF) and after both fermentation and distillation (PAFD) were provided by The Jerte Valley Cooperatives Group (Cáceres, Spain, 2022 campaign). The plum used to obtain plum brandy was crushed with rubber rollers, keeping the stone intact. The resulting intact stones, together with the pulp, were subjected to a biological must fermentation phase at a controlled temperature (18–20 °C) for 25–30 days. Subsequently, it was subjected to a distillation process using indirect heat in batch-loaded stills.

For the pretreatment of the plum pits, the experimental procedure proposed by Rodríguez-Blázquez et al. [6] was followed. Plum stones were air-dried at 40 °C (Digitheat oven, J.P Selecta®, Abrera, Barcelona, Spain) for 24 h and manually separated into their two components: shell and seed, using a hammer. The seeds were then crushed in an ultracentrifuge crusher (Retsh™ ZM200, Haan, Germany), and the particle size was reduced and homogenized to below 1 mm with a stainless-steel sieve.

### 3.3. Moisture Content of Seed Samples

Plum seeds moisture content was determined according to the standard procedure AOAC 925.10 [92], with some modifications. Approximately 2 g of seed sample was weighed on a dried crucible and introduced in an oven at 105 °C for 2 h and 30 min until a constant weight was obtained. After cooling, the crucible was weighed again, and the free water content was calculated as sample weight loss and expressed as a percentage in weight (mean ± standard deviation, n = 3).

### 3.4. Soxhlet Extraction of Plum Seeds

Oil extraction from plum seeds was performed following standard Soxhlet extraction [28] and the after-treatment described by Rodríguez-Blázquez et al. [6,12]. Approximately 15 g of plum seed with a particle size of less than 1 mm were extracted with 150 mL of *n*-hexane (seed-solvent ratio (1:10)) under reflux at 69 °C for 6 h (6–8 cycles per hour). Then, the solvent was evaporated using a rotary evaporator (Buchi™ Rotavapor™ R-100, Fisher Scientific, Hampton, VA, USA) at 69 °C. The collecting flask with the resulting plum oil and the by-product defatted plum seed contained in the cellulose cartridge were dried and placed in a vacuum oven (Vaciotem-TV, digital, J.P Selecta®, Abrera, Barcelona, Spain) at 40 °C for 24 h. This procedure was carried out in triplicate for each plum seed. Plum seed oils were stored in airtight, amber-colored glass bottles and kept at −20 °C, whereas the defatted plum seeds were stored in a clear plastic zip-lock bag until analysis. The percentages in weight (*w*, %) of plum seed oils and defatted plum seed by-products were expressed as mean ± standard deviation both on a dry basis.

### 3.5. Phenolic Extraction of Defatted Plum Seeds

With the aim of extracting phenolic compounds from defatted plum seed by-products, a sustainable MSPD method was used.

For this purpose, the optimized MSPD extraction previously developed was applied [12]. Briefly, 0.1000 g of defatted plum seeds and 0.1000 g of silica dispersant were mixed for 2 min to obtain a homogeneous powder and they were transferred to a 5 mL plastic tube. Then, 3 mL of EtOH-water (80:20 (*v/v*)) were added, and the mixture was stirred for 10 min at 402.48× *g*, using a vortex. A clear supernatant was collected after centrifugation for 30 min at 1528× *g* (centrifuge 5804, Eppendorf, Hamburg, Germany). Samples were prepared in triplicate.

### 3.6. Characterization of the Lipid Profile and Evaluation of the Health Lipid Indexes of Plum Seed Oils

The determination of fatty acid profile in the extracted plum seed oils was carried out using a gas chromatography (Agilent 6890) coupled to a Mass Spectrometer (Agilent 5973) following the internal derivatization procedure to fatty acid methyl esters (FAME). Prior to the chromatography analysis, the methyl esters of fatty acids were formed by methylation with sodium methoxide and esterification in an acid medium according to the usual methods described in the bibliography [93,94] with some modifications. In a 12 mL screw-capped tube, approximately 20 mg of seed oil were added with 0.6 mL of a methanolic internal standard solution (tridecanoic acid; 1730 mg·L$^{-1}$) and 1.4 mL of a methanolic sodium methoxide solution (4.3%, *w/v*). After stirring, the tube was kept for 10 min in an oven (Digitheat oven, J.P Selecta®, Abrera, Barcelona, Spain) at 60 °C. Then, 2 mL of a solution of sulfuric acid (4%, *v/v*) in methanol were added and again kept at 60 °C in an oven (Digitheat oven, J.P Selecta®, Abrera, Barcelona, Spain) for 20 min. For methyl esters extraction, 1 mL of distilled water and 1.5 mL of *n*-hexane were added. After shaking and centrifugation, 960 μL of *n*-hexane were recovered and placed in a 2 mL chromatography vial, followed by the addition of 40 μL of a solution of ethyl nonanoate (6400 mg·L$^{-1}$) in *n*-hexane, which was used as an internal standard. Finally, 1 μL was then injected into the gas chromatography system.

The chromatographic separation was carried out in a capillary chromatographic column (30 m length × 0.25 mm of internal dimeter × 0.25 μm film thickness) (DB-WAX-UI, J&W Scientific, Folsom, CA, USA). The column oven program was: 80 °C (5min), 8 °C/min from 80 °C to 145 °C, isothermal at 145 °C for 27 min, 4 °C /min up to 230 °C, and isothermal at 230 °C for 20 min. The injector temperature was 240 °C, and the injection was in split mode (1:40). The column flow was 1 mL·min$^{-1}$.

The mass spectrometer was operated in the electronic impact ionization mode (70 eV) and in SCAN (*m/z* range: 35–400) and SIM (*m/z* = 74, 87, 348, and 382) combined acquisition mode. Compound identification was carried out via a comparison of retention times and mass spectra of the reference compounds (commercial FAME mix and ethyl nonanoate) with those reported in the mass spectrum library NIST 2.0. The chromatogram of the FAME 37 SUPELCO standard mixture was registered and included in Figure S3. Moreover, the chromatograms of PBF, PAF, and PAFD seed oils are represented in Figure S4, Figure S5, and Figure S6, respectively. In addition, for quantification purposes, tridecanoic acid was used as an internal standard. Methyl esters integration peaks were carried out by using SIM chromatograms, and for ethyl nonanoate, *m/z* = 88 of the SCAN chromatograms. For obtaining the response factors (RF) of the methyl esters, the commercial mix was injected at three different concentrations (not diluted and diluted ½ and ¼), obtaining an average RF. That average RF was used to establish the percentage of each methyl ester in samples, and the results corresponding to the average RF with their respective standard deviation are shown in Table S2. It presented a variation among samples lower than 3.8%, thus indicating the correct derivatization process. The limit of detection (LOD) and quantification (LOQ) was determinate considering 3.3 times the background noise signal for the former and 10 times for the latter. Analyses were

performed in duplicate, and the composition of the plum seed oils was expressed as a percentage (%) considering the RF obtained for each fatty acid standard using Equation (1).

$$\frac{C_i^m}{C_t^m}\,(\%) = \frac{\frac{A_i^m}{R_{fi}}}{\sum_{i=1}^{n} \frac{A_i^m}{R_{fi}}} \times 100 \tag{1}$$

where $C_i^m$ is the concentration of each FAME in the sample, $C_t^m$ is the total concentration of FAMEs in the sample, $A_i^m$ is the individual area for each FAME, and $R_{fi}$ is the response factor obtained for each standard FAME.

According to the fatty acid profile obtained for each plum seed oil, the respective healthy nutritional value of each of them was determined. For this purpose, the total content of SFA, UFA, MUFA, and PUFA was determined, as well as the AI index, TI index, H/H ratio, and DFA index, as defined by Rodríguez-Blázquez et al. [6]. The following equations were used:

$$\sum SFA = C16:0 + C17:0 + C18:0 + C20:0 + C22:0 + C24:0 \tag{2}$$

$$\sum UFA = \sum MUFA + \sum PUFA \tag{3}$$

$$\sum MUFA = C16:1n7 + C18:1n7c + C18:1n9c \tag{4}$$

$$\sum PUFA = C18:2n6c + C18:3n3 + C20:1n9 \tag{5}$$

$$DFA = \sum MUFA + \sum PUFA + C18:0 \tag{6}$$

$$AI = \frac{C12:0 + 4(C14:0) + C16:0}{\sum MUFA + \sum PUFA} \tag{7}$$

$$H/H = \frac{C18:1 + \sum PUFA}{C14:0 + C16:0} \tag{8}$$

where C12:0 is dodecanoic acid, C14:0 is myristic acid, C16:0 corresponds to palmitic acid, C16:1n7 is palmitoleic acid, C17:0 is margaric acid, C18:0 is stearic acid, C18:1n7c is cis-vaccenic acid, C18:1n9c is oleic acid, C18:2n6c is linoleic acid, C18:3n3 is α-linolenic acid, C20:0 is arachidic acid, C20:1n9 is 11-eicosenoic acid, C22:0 is behenic acid, and C24:0 is lignoceric acid.

Finally, the O/L ratio was determined as another widely used parameter to establish the quality of plum seed oils [79].

### 3.7. Characterization of Tocopherols in Plum Seed Oils

The analysis of α-, β-, γ-, and δ-tocopherol was carried out via HPLC-PDA, following the procedure previously described by Aksoz et al. [10]. The equipment consisted of a Jasco LC-NetII/ADC degasser, a Jasco PU-2089 Plus quaternary gradient pump, and a Jasco Md-2018 Photodiode Array Detector. A Luna C18 column (5 μm 150 × 4.6 mm 100 Å, Phenomenex, Torrance, CA, USA) was capable of separating three out of four analytes, other than the β and γ homologs, using pure methanol as the mobile phase. The flow rate was set at 1 mL·min$^{-1}$, and the UV-vis detection was performed at both 292 and 305 nm. The identification of tocopherols was based on the comparison of retention time and spectral characteristics with those of the standards. Quantitative analyses were performed at 292 nm, given the higher signal sensitivity, using two external calibration curves, one at low concentrations (n = 5) between 0.5 and 20.0 mg·L$^{-1}$ for γ- and α-tocopherol and 0.3–20.0 mg·L$^{-1}$ for δ-tocopherol; and the other for high concentrations (n = 4), between 20.0 and 100 mg·L$^{-1}$. The estimation of the LOD and LOQ was achieved considering 3.3 times the background noise signal for the former and 10 times for the latter. The robustness of the method was studied at 10 and 100 mg·L$^{-1}$ for α, 5 and 50 mg·L$^{-1}$ for γ, and 2 and 50 mg·L$^{-1}$ for δ, via the injection of the same standard solution for each of the two calibration curves thrice (n = 3) during three consecutive days (N = 9). The precision results were calculated as an analysis of the peak area and the retention factor (Table 4).

Finally, for the analysis of plum seed oils, samples were filtered through a 0.45 μm nylon filter beforehand and diluted in a 1:3 ratio using 2-propanol as a solvent, following the protocol developed by Aksoz et al. [10]. Samples were analyzed in quadruplicate, and the results were expressed as mean ± standard deviation in mg per kg of dry plum seed sample. For illustrative purposes, Figure S2 represents the corresponding chromatogram for each sample, as well as the standard mixture used for reference.

### 3.8. Determination of Phenolic Compounds from Defatted Plum Seed Extracts

3.8.1. Spectrophotometric Methods

The total phenolic content of defatted plum seed extracts was determined using the Folin–Ciocalteu method [12]. After combining 750 μL of defatted plum seed extracts with 70 μL of the Folin–Ciocalteu reagent, 60 μL of 7.5 ($w/v$) of $Na_2CO_3$, and Milli-Q water to a final volume of 10 mL, the absorbance of the solution was measured at 720 nm (Thermo Scientific Multiskan spectrophotometer, Agilent Technologies, Santa Clara, CA, USA) and the obtained results were displayed in terms of mg of gallic acid equivalent per gram of defatted seed dried sample (mg GAE·g$^{-1}$ defatted seed), using gallic acid as standard (0–40 μM). The assay was performed in triplicate.

The total flavonoid content of defatted plum seed extracts was measured by the aluminum complexation colorimetry method [12]. Briefly, 750 μL of defatted plum seed extracts were combined with 2 mL of Milli-Q water and 150 μL of 5% ($w/v$) $NaNO_2$ for 5 min. Then, 150 μL of 10% ($w/v$) of $AlCl_3$ were added to the mixture and left for 5 min. After that, 1 mL of 1 M NaOH was added to stop the reaction solution and left for 15 min, which was then diluted to a volume of 10 mL with Milli-Q water. The absorbance of the flavonoid-Al (III) complex was measured at 415 nm, and the results were represented as mg of quercetin equivalent per gram of defatted seed dried sample (mg QE·g$^{-1}$ defatted seed) using quercetin as standard (0–45 μM).

3.8.2. Chromatographic Method

The individual polyphenol determination of the plum seed phenolic extract at the different stages of brandy processing was performed by HPLC-ESI-QTOF-MS, following the procedure previously described by Rodríguez-Blázquez et al. [12].

Agilent liquid chromatography system (Mod. 1200) was used consisting of a quaternary pump (G1311A), a coupled degasser (G1322A), an automatic injector with thermostat (G1367B), a column module with thermostat (G1316A), and a QTOF mass spectrometer (G1316A), with electrospray ionization (ESI) source at atmospheric pressure and JetStream technology, operating in negative mode and scanning (SCAN) mode in the range $m/z$ 100–1000. A capillary voltage of 4 kV and a pressure of 45 psi were employed in the chromatographic analyses. Nitrogen was used as the fogging and drying gas (10.0 L·min$^{-1}$, 325 °C), and the data treatment was performed using Masshunter Data Acquisition B.05.00, Masshunter Qualitative Analysis B.07.00, and Massprofinder Professional B.08.00.

The chromatographic separation was carried out using a Synergi™ C18 Fusion-RP 80 Å (150 mm, 3 mm I.D., 4 m, Phenomenex, Torrance, CA, USA), maintained at 30 °C. The flow rate was set to 0.5 mL·min$^{-1}$ and the injection volume to 20 μL. Moreover, a mobile phase consisting of 0.1% ($v/v$) formic acid (FA) aqueous solution (solvent A) and acetonitrile (solvent B) was used, operating in gradient elution as follows: 10% of solvent B was held for 0.1 min, it then increased linearly to 35% in 30 min, and achieved 70% in 5 min. This state was maintained for 2 min, after which a final linear increase to 90% B was obtained in 3 min and maintained for 5 min. Finally, the gradient was re-equilibrated. The identification and quantification of phenolic compounds present in defatted plum seed extracts were focused on high-resolution mass data collected from commercial standards and databases (FooDB and Mass Bank), and 5-level external calibration curves were obtained for the phenolic standards. When phenolic standards were not available, a semi-quantification was performed using the most similar standard available. In addition, the LOD and LOQ were determined considering the former is 3.3 times the background

noise signal and 10 for the latter. Finally, the results were represented in mg per gram of defatted dried seed. Figures S7–S23 represent the standard solution mass spectrum of gallic acid, 2,3-dihydroxibenzoic acid, *p*-coumaric acid, quercetin, kaempferol, caffeic acid, *trans*-ferulic acid, hesperidin, catechin, epicatechin, chlorogenic acid, kaempferol 3-rutinoside, neochlorogenic acid, vanillic acid, protocatechuate, syringic acid, amygdalin, and isorhamnetin 3-rutinoside. Figure S24 represents the corresponding chromatograms obtained for PBF, PAF, and PAFD samples.

### 3.9. Evaluation of Bioactive Properties of Plum Seed Oils and Defatted Plum Seed Extracts

3.9.1. Antioxidant Activity

**Free-radical scavenging assay:** The scavenging ability of plum seed oils and defatted seed by-products was evaluated against the DPPH.

On one hand, for the evaluation of the antioxidant capacity of plum seed oils, the method proposed by Rodríguez-Blázquez et al. [6] was followed. Briefly, in a 96-well microplate, 30 μL of eight methanolic working solutions (0–30 μL) made by diluting oil solutions in DMSO (448–850 mg·mL$^{-1}$) were combined with 270 μL of a $6 \cdot 10^{-5}$ M DPPH methanolic solution, following incubation for 60 min in the dark. Finally, the absorbance was measured at 515 nm. The results were represented as IC$_{50}$ values (mg·mL$^{-1}$ of oil), e.g., the concentration of sample needed to inhibit 50% of the original DPPH concentration, after the oil concentrations were plotted against the remaining DPPH percentages. Trolox was employed as a positive control. The experiment was run in duplicate, up for 22 days, to evaluate the oxidative stability of plum seed oils during storage. For this purpose, the data were fitted to first-order kinetics, and from this logarithmic calibration, the half-time was determined for each of the plum seed oils.

On the other hand, the antioxidant activity of the phenolic extracts was determined as described by Rodríguez-Blázquez et al. [6]. Concisely, 10–100 μL of sample aliquots and 100 μL of 0.28 mM DPPH solution in MeOH were combined to prepare a working solution in a 96-well microplate, with a total volume of 200 μL. Furthermore, a DPPH control and a blind control (defatted plum seeds mixed with pure MeOH) were prepared. Trolox served as a reference substance. The absorbance was measured at 515 nm after a 60 min incubation in the dark. Lastly, the IC$_{50}$ value was calculated and reported as milligrams of extract per gram of defatted dried seed. The assay was performed in three independent experiments.

**Lipid peroxidation assay:** The antioxidant activity of the phenolic extract of defatted plum seeds estimated as the capability to inhibit lipid peroxidation was evaluated via the in vitro TBARS method. The TBARS assay was conducted with porcine brain cell homogenates and sample extract at concentrations of 4.0–0.125 mg·mL$^{-1}$, in accordance with the protocol outlined by Gómez-Mejía et al. [77]. The inhibition ratio was calculated as the remaining percentage and represented in mg extract·g$^{-1}$ of defatted dried seed at the IC$_{50}$ value after the solutions' absorbance was measured at 532 nm. Every sample underwent three independent examinations. Furthermore, Trolox was employed as a positive control.

3.9.2. Anti-Amyloidogenic Activity

The anti-amyloidogenic effect was evaluated in the phenolic extract of PAFD defatted seed re-dissolved in HCl on pre-treated Aβ$_{42}$ monomer, employing Transmission electron microscopy (TEM) according to the procedure previously described by Vicente-Zurdo et al. [95], with slight modifications. Briefly, properly pre-treated Aβ$_{42}$ monomer was daily dissolved to obtain a final Aβ$_{42}$ working solution of 200 μM. In the aggregation experiments, a daily preparation of a 200 μM Fe(II) metal solution was conducted. Working solutions were prepared by diluting stock solutions in 10 mM HCl, including Aβ$_{42}$ alone, Aβ$_{42}$ + Fe(II), Aβ$_{42}$ + PAFD phenolic extract, and Aβ$_{42}$ + Fe(II) + PAFD phenolic extract. PAFD phenolic extracts were prepared and tested at concentrations of 25 and 50 mg·L$^{-1}$, while Aβ$_{42}$ and Fe(II) were added at 50 μM. To evaluate amyloid inhibition capacity, all working solutions were incubated at 37 °C for 48 h and subsequently analyzed using TEM.

Curcumin (50 μM) served as a negative control due to its recognized anti-amyloidogenic properties [96].

For TEM analysis, 1.5% (*w/v*) phosphotungstic acid was employed for negative staining of $A\beta_{42}$ fibrils. After removing the excess staining solution, the prepared grids were transferred and examined using a JEOL JEM 1400 Plus transmission electron microscope operating at 120 kV. TEM images were captured at different magnification powers. A comparative analysis was conducted to observe differences in the aggregation effect and the width of fibrils among the obtained images from the protein alone, in the presence of metal, in the presence of the target plum extracts, and in the presence of both. The fibril measurements were performed using the ImageJ software (https://imagej.net/ij/download.html) (n = 200).

### 3.9.3. Antibacterial Activity

The antibacterial activity of defatted plum seed phenolic extracts re-dissolved in water (0.156–20 mg·mL$^{-1}$) was tested against a Gram-negative bacteria (*Escherichia coli*) and a Gram-positive bacteria (*Staphylococcus aureus*) by the microdilution method combined with the *p*-iodonitrotetrazolium chloride (INT) rapid colorimetric assay proposed by Gómez-Mejía et al. [77]. Briefly, 100 μL of each diluted defatted plum seed extract (0.156–20 mg·mL$^{-1}$) were mixed with 100 μL Tryptic Soy Broth (TSB) and 10 μL of bacteria suspension in sterile water at $5.2 \times 10^8$ CFU·mL$^{-1}$. For each inoculum, Streptomycin (0.625–20 mg·L$^{-1}$) was used as a positive control in the TSB medium. In addition, three more controls were added: TSB medium inoculated with bacterial suspension, defatted seed phenolic extracts in TSB medium, and non-inoculated TSB medium. After incubation at 37 °C for 1 h, inhibition of bacterial growth was visually verified by color change after the addition of INT dye (50 μL at 0.2 mg·mL$^{-1}$). Antibacterial activity results were expressed as MIC; mg·mL$^{-1}$.

### *3.10. Statistical Analysis*

Data were statically analyzed by LSD multiple comparison test, ANOVA, multifactorial ANOVA, and PCA using the software package Statgraphics 19 (Statgraphics Technologies, Inc., Rockville, MD, USA).

## 4. Conclusions

In this work, formerly low-interest residues of plum seed employed in the plum brandy manufacturing process were explored, revealing a cascade valorization process to obtain high-value bioactive added compounds such as polyphenols, tocopherols, and unsaturated fatty acids. To establish the effect of processing in the seed samples and therefore, in the obtained compounds, a Soxhlet extraction using *n*-hexane as a solvent was effectively performed to separate antioxidant oils from defatted seed by-products. A sustainable and reliable MSPD extraction procedure was also employed to obtain phenolic extracts with bioactive potential.

As far as results go, it has not been observed that any of the brandy processing stages (before fermentation, after fermentation, and after both fermentation and distillation) positively affect the properties of the plum seeds, which showed that depending on the applicability that is sought, it may be of most interest to use the oil or the defatted seed. Considering the lipid fraction, in which the main components were C18:1n9c (72–75.56%) and C18:2n6c fatty acids (15.83–20%) with outstanding heart-healthy lipid indexes, particularly, PBF and PAF seed oils presented the greatest applicability in the nutraceutical, pharmaceutical and cosmetic industries, characterized by a high content of UFA, specifically C16:1n7 (0.765 ± 0.004%) and C17:1n7c (0.092 ± 0.001%), respectively. Both fatty acids contributed positively to the excellent oxidative stability (72–144 days for PBF and PAF seed oils, respectively) and to the antioxidant activity (IC$_{50}$ = 20–21 mg·mL$^{-1}$). Chiefly, PAF presented the lowest $\gamma/\alpha$ ratio (2.27 ± 0.02).

Regarding the hydrophilic fraction of the seeds, all defatted plum seed phenolic extracts (PBF, PAF, and PAFD) showed high antiradical activity ($IC_{50}$ = 0.9–1.9 mg·g$^{-1}$) and anti-lipid peroxidation activity ($IC_{50}$ = 1.3–5.0 mg·g$^{-1}$) with potential applicability in the nutraceutical, pharmaceutical or cosmetic industries. Albeit, in PBF and PAF phenolic extracts, it is necessary to remove amygdalin since it negatively affects their nutritional value. This stage is not necessary for the PAFD phenolic extract since amygdalin was not detected, which makes this extract stand out considerably from the others. In addition, it was the one that presented the greatest inhibition against *Escherichia coli* and *Staphylococcus aureus* growth (MIC = 20 mg·mL$^{-1}$), A$\beta_{42}$ aggregation (25 mg·mL$^{-1}$), and lipid peroxidation (1.3 mg·g$^{-1}$), with potential use in the nutraceutical industry or as an active ingredient in the manufacturing of food packaging.

**Supplementary Materials:** The following supporting information can be downloaded at https://www.mdpi.com/article/10.3390/ijms25021236/s1.

**Author Contributions:** Conceptualization, S.R.-B., L.P.-C., E.G.-M., D.V.-Z., N.R.-C., M.E.L.-G., J.J.R.-B. and R.M.; methodology, S.R.-B., L.P.-C., E.G.-M., D.V.-Z., N.R.-C., M.E.L.-G., J.J.R.-B. and R.M.; software, S.R.-B., L.P.-C., E.G.-M., D.V.-Z., N.R.-C., M.E.L.-G. and J.J.R.-B.; validation, S.R.-B., L.P.-C., E.G.-M., D.V.-Z., N.R.-C., M.E.L.-G., J.J.R.-B. and R.M.; formal analysis, S.R.-B., L.P.-C., E.G.-M., D.V.-Z. and J.J.R.-B.; investigation, S.R.-B., L.P.-C., E.G.-M., D.V.-Z., N.R.-C., M.E.L.-G., J.J.R.-B. and R.M.; resources, E.G.-M., N.R.-C., M.E.L.-G. and R.M.; data curation, S.R.-B., L.P.-C., E.G.-M., D.V.-Z. and J.J.R.-B.; writing—original draft preparation, S.R.-B., L.P.-C., D.V.-Z. and J.J.R.-B.; writing—review and editing, S.R.-B., L.P.-C., E.G.-M., D.V.-Z., N.R.-C., M.E.L.-G., J.J.R.-B. and R.M.; visualization, S.R.-B., L.P.-C., E.G.-M., D.V.-Z., N.R.-C., M.E.L.-G., J.J.R.-B. and R.M.; supervision, S.R.-B., L.P.-C., E.G.-M., D.V.-Z., N.R.-C., M.E.L.-G., J.J.R.-B. and R.M.; project administration, E.G.-M., N.R.-C., M.E.L.-G. and R.M.; funding acquisition, R.M. All authors have read and agreed to the published version of the manuscript.

**Funding:** This research has been funded by the Ministry of Science and Innovation, the state research agency, and by the European Union NextGenerationEU/ PRTR [project TED2021-129917B-I00]. Moreover, this work was supported and the Ministry of Science, Innovation and Universities [project PID 2020-114714RB-I00].

**Institutional Review Board Statement:** Not applicable.

**Informed Consent Statement:** Not applicable.

**Data Availability Statement:** The data presented in this study are available on request from the corresponding author.

**Acknowledgments:** The authors are grateful to the Analysis Service Unit facilities of ICTAN for the analysis of Liquid Chromatography and Mass Spectrometry and the sample providers, "The Jerte Valley Cooperatives Group". This work was supported by the Complutense University through a research staff contract in the "Investigo programme" (CT36/22-30-UCM-INV) and by the Ministry of Science and Innovation through a staff contract research support (PAII10/23-3/2023-05).

**Conflicts of Interest:** The authors declare no conflicts of interest.

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
