# Peer review of "The Potential of Plum Seed Residue: Unraveling the Effect of Processing on Phytochemical Composition and Bioactive Properties"

_ijms, doi:10.3390/ijms25021236_

Round 1

Reviewer 1 Report

Comments and Suggestions for Authors

The comments are as follows:

1. The main issue that has to be addressed is that manuscript is too descriptive. The whole text should be revised and authors should focus on the relevant data. It is necessary to follow he aim of the study and comment on the results accordingly.

2. The authors should choose if they will use term "plum" or "Prunus domestica" throughout the text, not both, especially not together.

3. Line 178-182. It is not necessary to repeat the values from Table 2.

4. Line 409-414. This part should be deleted because it represents a repetition of the text.

5. Line 453-456. This part should be deleted as well. Well-known fact.

6. It is not necessary to repeat values from tables and figures in the text. The authors should focus just on the relevant data.

7. Subsection 3.2. Moisture content for the dried sample material is missing.

8. Line 995-997. This part should be deleted. Not relevant for this section.

9. Abbreviations should be defined just the first time they are mentioned in the text and used as such throughout the manuscript.

10. Subsection 3.7.1. Calibration curve and correlation factor for TPC and TFC are missing.

11. Six self-citations of the co-authors Gómez-Mejía, Rosales-Conrado and León-González should be checked and reduced.

12. The text should be checked for typos and grammatical errors.

Author Response

Dear Reviewer,

The authors of the manuscript are grateful for you to taking time to review it.  Please find the detailed responses point-by-point below and the corresponding corrections highlighted or in track changes in the re-summited files.

Comment 1: The main issue that has to be addressed is that manuscript is too descriptive. The whole text should be revised and authors should focus on the relevant data. It is necessary to follow he aim of the study and comment on the results accordingly.

Response 1: The authors have adequately followed their proposal. The entire text has been reviewed and checked, the aim of the work has been followed, and all the data considered relevant to the study have been adequately explained. Likewise, repetitive information has been eliminated and the descriptive explanation of the results has been reduced. However, the authors consider it necessary to maintain certain descriptive information that can help the reader to understand how the effect of the processing that plum seeds undergo can affect the compositions and profiles of bioactive compounds. Suchlike, in sections: 2.5. Evaluation of bioactive properties of plum seed oils and defatted plum seeds and 2.6. Multivariate statistical analysis, information that has been explained more briefly in other sections has been grouped, since the authors consider it necessary to return to these explanations in greater detail to give an overview of the work.

Comment 2: The authors should choose if they will use term "plum" or "Prunus domestica" throughout the text, not both, especially not together.

Response 2: The authors have revised the text and have decided to use the term plum throughout the manuscript. However, they consider that it is necessary to also indicate the variety P. domestica L. in the materials and methods of the raw materials subsection to fully specify the variety. In addition, when referring to bibliographic data in the introduction or also in the results and discussion section, it has been decided to indicate only the term P. domestica L. since in these cases, the plum variety is crucial to be specified.

Comment 3: Line 178-182. It is not necessary to repeat the values from Table 2.

Response 3: These lines of text have been checked, and all repeated values referring to Table 2 have been removed.

Comment 4: Line 409-414. This part should be deleted because it represents a repetition of the text.

Response 4: The part of the text indicated in this comment has been deleted.

Comment 5: Line 453-456. This part should be deleted as well. Well-known fact.

Response 5: This part of the text has also been deleted.

Comment 6: It is not necessary to repeat values from tables and figures in the text. The authors should focus just on the relevant data.

Response 6: The authors have addressed the issue raised and in the revised text the repeated values from tables and figures have been deleted. Notwithstanding, the data values in Figures 1 and 2 have been maintained since the authors consider it necessary to indicate the exact values since the bar graphs do not show them properly.

Comment 7: Subsection 3.2. Moisture content for the dried sample material is missing.

Response 7: The text has been checked and section 3.3. Moisture content of seed samples has been added.

Comment 8: Line 995-997. This part should be deleted. Not relevant for this section.

Response 8: the authors have followed the above comment and removed these lines from the text.

Comment 9: Abbreviations should be defined just the first time they are mentioned in the text and used as such throughout the manuscript.

Response 9: The whole text and abbreviations have been checked, hence, they will be only explained the first time they appear and not throughout the rest of the manuscript.

Comment 10: Subsection 3.7.1. Calibration curve and correlation factor for TPC and TFC are missing.

Response 10: Calibration curves for total polyphenol content (TPC) and for total flavonoid content (TFC) have been added in Figure 2 caption, as well as their corresponding correlation factors. Additionally, the experimental procedure followed for the determination of TPC and TFC in defatted seed samples can be found in subsection 3.8.1. Spectrophotometric methods from section 3.8. Determination of phenolic compounds from defatted plum seed extracts, as well as the determination of standards calibration curve procedure.

Comment 11: Six self-citations of the co-authors Gómez-Mejía, Rosales-Conrado and León-González should be checked and reduced.

Response 11: According to the reviewer’s point, the literature have been reviewed and reduced in the resubmitted manuscript, remaining in those cases where it is strictly necessary (reference 6, 12, 78 and 86).

Comment 12: The text should be checked for typos and grammatical errors.

Response 12: The authors are grateful for the reviewer’s suggestion and have consequntly checked the entire text for typos and grammatical errors, which have been corrected.

Reviewer 2 Report

Comments and Suggestions for Authors

Kindly reply to the following comments:

1-      Result section: Add Discussion in the title

2-      Use P. domestica instead of Prunus domestica

3-      Line 154 to 165: move to material and methods section

4-      Kindly send Chromatograms of plum (Prunus domestica L.) seed oil after fermentation and after both fermentation and distillation

5-      Table S2: Identification of phenolic compounds by high-resolution liquid chromatography coupled to a quadrupole time-of-flight mass spectrometer (HPLC-ESI-QTOF-MS) in standards solution: kindly give more explanation, kindly give more explanation?

6-      Kindly provide chromatograms related to HPLC-ESI-QTOF-MS of the three samples? Please add as supplementary data.

7-      Precise the standard that you used for antibacterial activity in table 9.

8-      Line 1250: and lipid peroxidation (),: please add the missing information

Comments on the Quality of English Language

 Minor editing of English language required

Author Response

Dear Reviewer,

Thank you for taking the time to revise this manuscript. Please find the detailed responses below and the corresponding revisions/corrections highlighted/in track changes in the re-submitted files.

Comment 1: Result section: Add Discussion in the title

Response 1: According to the reviewer’s suggestion,“Discussion“ has been added to the title of section 2. Results and discussion.

Comment 2: Use P. domestica instead of Prunus domestica

Response 2: In line with the preceding note, the term Prunus domestica L. has been simplified to P. domestica L. throughout the text of the revised manuscript, except when describing the raw material and the first description, except where it is considered most essential to specify the full name, i.e. the first time it is cited and in section 3.2. Raw materials.

Comment 3: Line 154 to 165: move to material and methods section

Response 3: Lines 154-165 have been moved to the materials and methods section. In the resubmitted manuscript they can be found on lines 1020-1025.

Comment 4: Kindly send Chromatograms of plum (Prunus domestica L.) seed oil after fermentation and after both fermentation and distillation

Response 4: The chromatograms corresponding to plum seed oils after fermentation and after fermentation and distillation have been included in the supplementary material, specifically Figure S5 and Figure S6. In addition, a reference to these figures in the manuscript has been added (lines 1020-1025).

Comment 5: Table S2: Identification of phenolic compounds by high-resolution liquid chromatography coupled to a quadrupole time-of-flight mass spectrometer (HPLC-ESI-QTOF-MS) in standards solution: kindly give more explanation, kindly give more explanation?

Response 5: According to the reviewer’s suggestion more information has been added to Table S1:  Identification of phenolic compounds by high-resolution liquid chromatography coupled to a quadrupole time-of-flight mass spectrometer (HPLC-ESI-QTOF-MS) in plum seed phenolic extracts, specifically in lines 438-445 of the resubmitted manuscript.

Comment 6: Kindly provide chromatograms related to HPLC-ESI-QTOF-MS of the three samples? Please add as supplementary data.

Response 6: The requested chromatograms have been added in supplementary materials (Figure S24). Furthermore, the authors have found it appropriate to add the SIM Scan mode chromatograms of the standards of those polyphenols found in the studied samples (Figures S7-S24). A reference to these chromatograms in the text of the resubmitted manuscript has been added (lines 1030-1035).

Comment 7: Precise the standard that you used for antibacterial activity in table 9.

Response 7: The standard used for antimicrobial activity has been indicated in Table 9 (Streptomycin antibiotic positive standard). Furthermore, it has also been indicated in section 3.1. Reagents, standards, bacterial strains and solvents.

Comment 8: Line 1250: and lipid peroxidation (),: please add the missing information

Response 8: The lines indicated have been revised and the missing information has been added (lines 1238-1239).

Comments on the Quality of English Language: Minor editing of English language required

Response: The entire text of the manuscript has been checked, and all the mistakes corresponding to the quality of the English Language have been amended.

Reviewer 3 Report

Comments and Suggestions for Authors

The study “The potential of plum seed residue: unraveling the effect of processing on phytochemical composition and bioactive properties” analyzed bioactive compounds in plum (Prunus domestica L.) seeds across brandy production stages. Oils extracted using n-hexane were rich in heart-healthy unsaturated fatty acids, mainly oleic acid, and tocopherols with antioxidant properties. Defatted seed extracts showed antimicrobial and antioxidant activities, attributed to compounds like neochlorogenic acid and caffeic acid. Principal Component Analysis correlated these findings with brandy processing stages, identifying the defatted plum seed extract post-fermentation and distillation as the most suitable for pharmaceutical, nutraceutical, and cosmetic applications. The work is interesting, but I think it can be improved by answering the following questions:

1.       Given your findings on the fatty acid profile changes in plum seed oils during brandy production processes, have you investigated or do you plan to investigate any potential alterations in the sensory qualities (like flavor, aroma, and texture) of these oils? If so, how might these changes impact the potential culinary or industrial applications of the oils?

2.       By using DPPH and TBARS assays, your work demonstrated that the antioxidant activity of the defatted plum seeds varied depending on the stage of processing. Could you talk about how these findings might affect the creation of functional food products? More specifically, what impact can the differences in antioxidant activities have on the nutritional content and use of these defatted seeds in the food and cosmetic sectors?

3.       Your study indicates that the antimicrobial activity against Escherichia coli and Staphylococcus aureus varies among the different plum seed extracts (PBF, PAF, and PAFD). Could you elaborate on the specific phenolic compounds or combinations thereof that contribute most significantly to the observed antimicrobial activities? Additionally, how do the changes in phenolic composition during the processing stages influence the effectiveness of these extracts against different types of bacteria?"

4.       How might the utilization of PAF and PAFD defatted seeds be adjusted to maximize safety and efficacy, given their potential application as antimicrobial agents in food packaging and agri-food ingredients? What additional difficulties and factors need to be taken into account when using these defatted seed extracts in actual food packaging and preservation systems?

5.       You discovered a negative link with oxidative stability and a high positive correlation with certain fatty acids and plum seed oils' antioxidant potential in your PCA analysis. Could you explain how these results might influence plum varieties' selective breeding or genetic alteration to improve the bioactive qualities of their seeds for certain industrial uses, as in cosmetics or nutraceuticals?

6.       Your PCA analysis revealed various phenolic profiles and their related bioactive characteristics with respect to the defatted plum seed extracts. In what ways might this knowledge be applied to the focused creation of active packaging materials or functional food products? Furthermore, what are the possible ramifications and approaches for enhancing the processing parameters of plum seeds to optimize their advantageous phenolic content and bioactivity for these purposes?

Author Response

Dear Reviewer,

The authors appreciate your time spent reviewing the manuscript. Please find the detailed responses point-by-point below and the corresponding corrections highlighted or in track changes in the re-summited files.

Comment 1: Given your findings on the fatty acid profile changes in plum seed oils during brandy production processes, have you investigated or do you plan to investigate any potential alterations in the sensory qualities (like flavor, aroma, and texture) of these oils? If so, how might these changes impact the potential culinary or industrial applications of the oils?

Response 1: The authors thank the reviewer for the interesting suggestion, although no exploration has been done on the sensory qualities of the oil that you kindly mentioned, and how this could affect its culinary applications, given that it was not within the aims of this present work, even though it could be considered for the near future. Having said that, the authors expect that the organoleptic properties of the oils will be modified during the brandy processing. Further to this, information on the observed changes in the coloration of plum oils has been added in supplementary materials (Figure S1), and explained in lines 131-134 of the manuscript. In addition, information has been added about the observed changes corresponding to the aroma of the oil.

Comment 2: By using DPPH and TBARS assays, your work demonstrated that the antioxidant activity of the defatted plum seeds varied depending on the stage of processing. Could you talk about how these findings might affect the creation of functional food products? More specifically, what impact can the differences in antioxidant activities have on the nutritional content and use of these defatted seeds in the food and cosmetic sectors?

Response 2: Thank you for your comment. The authors have added information on how differences in the antioxidant activities of phenolic extracts from defatted seeds may impact their use in nutraceutical, agri-food or cosmetic sectors. Briefly, it is indicated that the outstanding TBARS antioxidant activity of seed extracts could be of interest in agri-food industry for active packaging elaboration to protect food from external oxidizing agents. Concerning the high antioxidant capacity DPPH of the seed extracts, could be interesting for them to be applied in cosmetic industry as an active ingredient in the development of anti-aging cosmetics or anti-stain, among others. However, it is important to point out that, depending on the product to be developed and the sector of application, the most suitable seed extract for these purposes will have to be the one already selected. It is also crucial to consider other factors of interest in the extracts under study, such as phenolic content and antimicrobial activities.

Comment 3: Your study indicates that the antimicrobial activity against Escherichia coli and Staphylococcus aureus varies among the different plum seed extracts (PBF, PAF, and PAFD). Could you elaborate on the specific phenolic compounds or combinations thereof that contribute most significantly to the observed antimicrobial activities? Additionally, how do the changes in phenolic composition during the processing stages influence the effectiveness of these extracts against different types of bacteria?”

Response 3: To study the effect of the phenolic composition of PBF, PAF and PAFD extracts on their respective antimicrobial activities, a Principal Component Analysis (PCA) has been performed as shown in Figure 6. In addition, information has been added on which phenolic compounds are most effective against Escherichia coli and Staphylococcus aureus bacteria in lines 874-882.

Comment 4: How might the utilization of PAF and PAFD defatted seeds be adjusted to maximize safety and efficacy, given their potential application as antimicrobial agents in food packaging and agri-food ingredients? What additional difficulties and factors need to be taken into account when using these defatted seed extracts in actual food packaging and preservation systems?

Response 4: To ensure the safety and effectiveness of PAF and PAFD seed extracts as potential antimicrobial agents in the formulation of active packaging, it is first necessary to apply method for the elimination of amygdalin in PAF extract, as well as to carry out subsequent studies to ensure that amygdalin concentration levels are below those considered harmful to health. Although this cyanogenic glucoside has not been detected in PAFD extract using the chromatographic method that we have performed, it is required to carry out new safety controls to ensure its effectiveness. Furthermore, in the PCA of Figure 6 it has been seen that the antimicrobial activities against E. coli and S. aureus bacteria are highly correlated with the phenolic content. It would be interesting and required to apply stability studies of the antioxidant activity of DPPH and TBARS during storage time to ensure that there are no possible changes in the phenolic content, which could negatively affect their antimicrobial activity and, thus, reduce their efficacy as active packaging. It is also noteworthy that for the possible application of the studied phenolic extracts as additives in the formulation of active packaging, one difficulty that needs to be addressed is their compatibility with other lignocellulosic materials such as hemicellulose or lignin (among others) currently used in this industry.

Comment 5: You discovered a negative link with oxidative stability and a high positive correlation with certain fatty acids and plum seed oils' antioxidant potential in your PCA analysis. Could you explain how these results might influence plum varieties' selective breeding or genetic alteration to improve the bioactive qualities of their seeds for certain industrial uses, as in cosmetics or nutraceuticals?

Response 5: Sadly, in our study, we have solely studied the Prunus domestica L. variety, hence, to know if these results could be extrapolated to other plum varieties, new studies would have to be carried out. Even if any data was available, it could not be possible to make a reliable extrapolation of our results.

Comment 6: Your PCA analysis revealed various phenolic profiles and their related bioactive characteristics with respect to the defatted plum seed extracts. In what ways might this knowledge be applied to the focused creation of active packaging materials or functional food products? Furthermore, what are the possible ramifications and approaches for enhancing the processing parameters of plum seeds to optimize their advantageous phenolic content and bioactivity for these purposes?

Response 6: Information corresponding to the impact that variations in the phenolic profile of the extracts (Figure 6) may have on the formulation of active packaging has been added, specifically in lines 874-882. Regarding approaches to improve the processing parameters of plum seeds to optimize their advantageous phenolic content and bioactivity for these purposes, it would be necessary to carry out new optimization studies, where various fermentation temperatures as well as different distillation temperatures were studied to statistically select the optimal temperature in both cases, in which the seed extracts present the best bioactive properties. However, in this work, the plum pit residues obtained in the different stages of brandy processing have been supplied by the Valle del Jerte cooperative, so we have not had the opportunity to study these effects as they had already suffered the fermentation and distillation processes.

Reviewer 4 Report

Comments and Suggestions for Authors

Overall the manuscript is well written and the experiments are well executed. The author needs to improve the abstract with a line of more significant outcomes and future prospects of the current research. The author should also elaborate on the introduction part. In the section 2. heading the author has mentioned only "result", it should be mentioned as "results and discussion". Besides the author should cite more recent references in the discussion part. 

Author Response

Dear Reviewer,

Thank you very much for taking the time to review this manuscript. Please find the detailed responses below and the corresponding revisions/corrections highlighted/in track changes in the re-submitted files. Specifically, the abstract has been revised and improved, highlighting at the end future applications of interest that defatted plum oils and seeds could have. Furthermore, in the introduction, specifically lines 95-111, the prospects and applications that these waste materials could present have been indicated. To add on, the title of section 2 has been changed to Results and discussion. Additionally, the authors have updated several references in a bid to ensure that our results were in accordance with the most up-to-date findings in the field. However, it should be noted that certain references from over 10 years ago have occasionally have been retain as thoroughly appropriate for the situation in question, either because there are no alternatives available at present (reference 46), or because they referred to well-established official methods of analysis (references 93 and 94).

Round 2

Reviewer 4 Report

Comments and Suggestions for Authors

The author has addressed the issues raised and revised the manuscript accordingly.

Comments on the Quality of English Language

Minor editing of English language required